# LoRA Provably Reduces Forgetting and Enables Adapter Merging in Multiclass Linear Classification

## Abstract

Low–Rank Adaptation (LoRA) has become the de-facto parameter-efficient fine-tuning algorithm. Besides training-efficiency, practitioners observe two striking benefits: *(i)* remarkable resistance to *catastrophic forgetting*, and *(ii)* independently trained adapters can be *merged* into a single model that performs well on multiple tasks. Despite their practical importance, these phenomena have lacked rigorous theoretical explanation. In this work, we provide the first theoretical justification for the aforementioned phenomena by analyzing the structure of LoRA solutions in multiclass linear classification problems for orthogonal tasks. Our analysis shows that, under suitable weight regularization, the optimal LoRA adapter aligns exactly with the *max-margin* (hard-margin SVM) solution for the fine-tuning data. This alignment lets us track in closed form how the normalized margins on the pre-training data, fine-tuning data and their union vary with the regularization parameter. For *(i)*, we observe a trade-off: decreasing the regularization parameter enlarges the fine-tuning margin while proportionally shrinking the pre-training margin, never collapsing it to zero. Concerning *(ii)*, we view the merged weights through the same margin lens, we prove why merging succeeds and derive optimal mixing coefficients that maximize the margin on the union of all tasks. Finally, we numerically validate our theory across multiple deep learning architectures and task configurations. The empirical results closely match our theoretical predictions. Taken together, our results give the first principled explanation for LoRA's resistance to forgetting and its surprising merging ability.

## 1 Introduction

Foundation models like GPT-4, Gemini, and Deepseek-v3 have revolutionized Artificial Intelligence (AI) capabilities across numerous domains (OpenAI, 2023; Team et al., 2023; Liu et al., 2024). However, deploying these models in real-world applications typically requires fine-tuning on specialized datasets to meet accuracy, safety, and alignment requirements. Traditional full fine-tuning, which requires optimizing over all parameters of the pre-trained model, presents prohibitive computational barriers, requiring massive memory footprints, extended training times, and substantial storage resources. Low-Rank Adaptation (LoRA) (Hu et al., 2022) has emerged as a breakthrough solution, augmenting pre-trained weight matrices $W$ with low-rank adapters $B, A$ while keeping $W$ frozen during fine-tuning. Empirically, LoRA achieves accuracy comparable to full fine-tuning while training less than one percent of the original parameters. While LoRA's primary appeal was initially its efficiency, practitioners have discovered two remarkable additional benefits that lack a formal explanation:

**Resistance to catastrophic forgetting.** In fine-tuning and continual learning, catastrophic forgetting, i.e., the performance degradation on previously learned tasks after adapting to new ones, represents a fundamental challenge (Kirkpatrick et al., 2017; McCloskey & Cohen, 1989; Ramasesh et al., 2021). Surprisingly, recent empirical studies have consistently shown that LoRA exhibits a strong resistance to catastrophic forgetting, retaining prior knowledge even after extensive adaptation (Biderman et al., 2024; Qiao & Mahdavi, 2024; Wistuba et al., 2023). This unexpected robustness has made LoRA particularly valuable for incremental adaptation scenarios, yet without theoretical understanding of why this occurs.

**Effective adapter merging.** In the context of learning multiple tasks, maintaining separate fine-tuned models for each task requires substantial storage. Remarkably, studies have shown that LoRA adapters independently trained on distinct tasks can be directly merged through simple weighted summation of adapter weights into a single unified adapter that maintains high performance across all original tasks (Huang et al., 2023; Yadav et al., 2023; Wang et al., 2024; Zhao et al., 2024; Yu et al., 2024). This property enables extraordinary flexibility in model deployment and management, but lacks principled explanation for its effectiveness.

Despite these empirical benefits, the theoretical mechanisms underlying LoRA's resistance to *catastrophic forgetting* and its *adapter merging* capabilities have remained elusive. In this work, we take the first step toward closing this gap by providing theoretical explanations for these phenomena.

**Paper contributions.** In this work, we characterize the global minimizer of LoRA in the context of multiclass linear classification problems, focusing on an *orthogonal-tasks* regime where each fine-tuning dataset is orthogonal to both the pre-training data and other fine-tuning tasks. Compared with prior works on theoretical analysis of LoRA (Please see Appendix B for a detailed literature review), our contributions are as follows:

- *Characterization of optimal solution:* We provide the first complete characterization of LoRA's global minimizer across different regularization regimes (Theorem 3.1). *(i) high-penalty regime:* when the regularization parameter is large, LoRA adapters learn nothing, and equal zero at the global minimum. *(ii) intermediate regime:* when the regularization parameter is moderate, LoRA adapters align with the hard–margin SVM direction on the fine-tuning data. *(iii) low-penalty regime:* when the regularization parameter is small, LoRA adapters align with a non-max-margin classifier whose direction generally does not have a clear closed-form expression. Nevertheless, as the regularization parameter approaches zero, we show that its direction converges to a simple and interpretable structure.

- *Theory for reduced forgetting:* We define *forgetting* as the reduction in margin that the fine-tuned model exhibits on the pre-training data, and derive closed-form expressions for normalized margins on pre-training data, fine-tuning data and their union (Theorem 3.2). We theoretically characterize that LoRA's margins are governed by the regularization parameter, and observe the following trade-off: smaller regularization parameter leads to larger Frobenius-norm ratio of the adapter to the pre-trained weights, which leads to more *forgetting*. At the same time, the margin on the fine-tuning task increases. Finally, we identify the regularization parameter that maximizes the margin of the union of pre-training and fine-tuning tasks, optimally balancing retention of the old task with performance on the new one.

- *Theoretical foundation for adapter merging:* We derive closed-form expressions for the merged model's margin on each task under the assumption that the level of regularization for each task lies in the *intermediate regime* (see Theorem 3.3). We show that the margin of the unified model on each task remains strictly positive, and this result explains why *adapter merging* works in theory. Moreover, we prove that properly chosen mixing coefficients, which can be obtained in closed form, maximize the margin of the merged model on the union of *all* tasks.

- *Numerical validations:* We complement our theoretical analysis with empirical evaluations on real-world datasets using modern deep learning architectures. Our experiments confirm the following: *(i)* the existence of an optimal regularization level that maximizes the performance of the fine-tuned model, with our theoretically derived value matching the empirical optimum, and *(ii)* the effectiveness of our closed-form mixing coefficients for adapter merging.

## 2 PRELIMINARIES

In this section, we begin by introducing the multiclass linear classification problem and the corresponding hard-margin SVM formulation. We then describe the specific problem setup considered in this work, which consists of a pre-training and fine-tuning phase. Finally, we outline the key assumptions that underpin our analysis, and discuss their motivation and implications.

## 2.1 BACKGROUND ON MULTICLASS LINEAR CLASSIFICATION

We begin by reviewing the standard $K$-class linear classification problem, which serves as the foundation for our analysis. Given a dataset $\mathcal{D}$, the goal is to learn a weight matrix $W \in \mathbb{R}^{K \times d}$ that minimizes the empirical cross-entropy loss:

$$\min_{W \in \mathbb{R}^{K \times d}} L(W; \mathcal{D}) := \sum_{(\boldsymbol{x}, \boldsymbol{y}) \in \mathcal{D}} \mathcal{L}_{\mathrm{CE}}(\boldsymbol{y}, W\boldsymbol{x}), \tag{1}$$

where $\boldsymbol{x} \in \mathbb{R}^d$ is the input and $\boldsymbol{y} \in \mathbb{R}^K$ is a one-hot label vector. The cross-entropy loss is given by:

$$\mathcal{L}_{\mathrm{CE}}(\boldsymbol{y}, \hat{\boldsymbol{y}}) := -\sum_{c=1}^{K} y_c \cdot \log(z_c), \quad \text{where} \quad z_c = \frac{\exp(\hat{y}_c)}{\sum_{i=1}^{K} \exp(\hat{y}_i)}. \tag{2}$$

Here, $\hat{\boldsymbol{y}} = W\boldsymbol{x}$ represents the class logits, and $z_c$ is the softmax probability assigned to class $c$.

Recent studies have shown that when data is linearly separable, gradient descent (GD) implicitly biases the solution of (1) toward the max-margin classifier (Lyu & Li, 2019; Soudry et al., 2018; Gunasekar et al., 2018; Frei et al., 2022; Ravi et al., 2024). More specifically, the limiting solution (as the iteration of GD tends to infinity) satisfies the following hard-margin SVM problem:

$$\min_{W \in \mathbb{R}^{K \times d}} \|W\|_F^2 \quad \text{subject to} \quad C(W; \mathcal{D}) \geq 1, \tag{3}$$

where $C(W; \mathcal{D})$ is the minimum margin over the dataset:

$$C(W; \mathcal{D}) := \min_{(\boldsymbol{x}, \boldsymbol{y}) \in \mathcal{D}} \min_{k \neq c, y_c = 1} \left( \boldsymbol{w}_c^\top \boldsymbol{x} - \boldsymbol{w}_k^\top \boldsymbol{x} \right), \quad \text{with} \quad W = [\boldsymbol{w}_1; \ldots; \boldsymbol{w}_K]^\top.$$

The Hard-margin SVM problem in (3) seeks a weight matrix with minimal norm that separates all examples with at least unit margin between the correct class and the nearest competing class.

Intuitively, classification decisions depend primarily on the direction of the weight matrix rather than its scale. To enable comparisons across different scales of $W$, we define the *normalized margin*:

$$\gamma(W; \mathcal{D}) := \frac{C(W; \mathcal{D})}{\|W\|_F}. \tag{4}$$

While margin is well understood in binary classification ($K = 2$), there is no universally accepted definition in the multiclass setting. We adopt the formulation of Crammer & Singer (2001), which has been used in recent theoretical studies (Lyu & Li, 2019; Ravi et al., 2024).

## 2.2 PRE-TRAINING AND FINE-TUNING SETUP

We now describe the specific pre-training and fine-tuning framework that is the focus of this work. This setup is common in practice and serves as the foundation for our theoretical analysis.

**Pre-training stage.** Let $\mathcal{D}_{\mathrm{pre}}$ be a labeled dataset with $K$ classes. We assume a pre-trained linear classifier $W_{\mathrm{pre}} \in \mathbb{R}^{K \times d}$ has been obtained by minimizing the cross-entropy loss:

$$W_{\mathrm{pre}} \in \arg\min_{W \in \mathbb{R}^{K \times d}} L(W; \mathcal{D}_{\mathrm{pre}}). \tag{5}$$

**Fine-tuning stage.** In the fine-tuning stage, we aim to adapt the pre-trained model to a downstream task with dataset $\mathcal{D}_{\mathrm{ft}}$, which contains $\bar{K} \leq K$ new classes, each with $n$ samples. This pre-training and fine-tuning setup, where the fine-tuning stage involves new samples that are a subset of the classes present during pre-training, reflects several practical scenarios. These include *domain-incremental learning* (Esaki et al., 2024), *domain shift adaptation* (Zohrizadeh et al., 2019; Zhang et al., 2022), and *bias-rebalancing fine-tuning* (Li & Xu, 2021; ValizadehAslani et al., 2024).

When LoRA is applied to adapt the pre-trained model to the fine-tuning task, we introduce a low-rank update to the weights, parameterized by matrices $B \in \mathbb{R}^{K \times r}$ and $A \in \mathbb{R}^{r \times d}$:

$$\min_{B, A} L(W_{\mathrm{pre}} + BA; \mathcal{D}_{\mathrm{ft}}) + \frac{\lambda}{2}(\|B\|_F^2 + \|A\|_F^2). \tag{6}$$

Here, the Frobenius-norm penalties explicitly constrain the adapters in weight space, limiting the deviation of the fine-tuned model from the pretrained initialization. Such regularization has been adopted in prior empirical studies (Hu et al., 2022; Biderman et al., 2024; Wistuba et al., 2023), making it a natural and widely used variant of the standard LoRA formulation.

Our goal is to understand the structure of the optimal solution to (6), and how it explains LoRA's resistence to catastrophic forgetting, and effectiveness in enabling adapter merging.

## 2.3 ASSUMPTIONS AND THEIR IMPLICATIONS

Throughout the paper, we make the following assumptions.

**Assumption 2.1.** *The input data dimension is larger than the total number of classes, i.e., $d \geq K$.*

**Assumption 2.2.** *The rank of the LoRA adapters is larger than or equal to the number of classes in the fine-tuning dataset, i.e., $r \geq \bar{K}$.*

**Assumption 2.3.** *The combined pre-training and fine-tuning datasets are linearly separable. Every fine-tuning feature vector has unit Euclidean norm, is orthogonal to every pre-training feature vector, and is also orthogonal to every other fine-tuning feature vector, i.e.,*

$$(a) \quad \|\bar{\boldsymbol{x}}\| = 1 \,, \bar{\boldsymbol{x}}^\top \bar{\boldsymbol{x}}' = 0 \,, \quad \forall \bar{\boldsymbol{x}}, \bar{\boldsymbol{x}}' \in \mathcal{D}_{\mathrm{ft}} \,, \qquad (b) \quad \bar{\boldsymbol{x}}^\top \boldsymbol{x} = 0 \,, \quad \forall \boldsymbol{x} \in \mathcal{D}_{\mathrm{pre}}, \forall \bar{\boldsymbol{x}} \in \mathcal{D}_{\mathrm{ft}} \,.$$

**Assumption 2.4.** *The pre-trained classifier $W_{\mathrm{pre}}$ is a scaled version of the hard-margin SVM solution on $\mathcal{D}_{\mathrm{pre}}$: $W_{\mathrm{pre}} = \rho_{\mathrm{pre}} \cdot W_{\mathrm{pre}}^{\mathrm{SVM}}/\|W_{\mathrm{pre}}^{\mathrm{SVM}}\|_F$, for some scalar $\rho_{\mathrm{pre}} > 0$, where $W_{\mathrm{pre}}^{\mathrm{SVM}}$ is the unique solution to:*

$$W_{\mathrm{pre}}^{\mathrm{SVM}} = \arg\min_{W \in \mathbb{R}^{K \times d}} \tfrac{1}{2}\|W\|_F^2 \quad \textit{subject to} \quad C(W; \mathcal{D}_{\mathrm{pre}}) \geq 1 \,.$$

We now briefly justify our assumptions and illustrate how they represent valid simplifications of real-world scenarios, preserving essential characteristics needed for theoretical analysis.

*Assumption 2.1* is mild since for most machine learning tasks, the input dimension is greatly larger than the number of classes, i.e., $d = 3072 > 100 = K$ in CIFAR-100 (Krizhevsky et al., 2009). *Assumption 2.2* requires only $r \geq \bar{K}$, meaning the LoRA rank needs only to exceed the number of new classes, and is independent of the number of samples, which aligns with practical implementations where LoRA with rank ranging from 8 to 64 successfully handles tasks with thousands of samples per class. Additionally, the orthogonality condition in *Assumption 2.3*, while restrictive, is common in theoretical analyses (Frei et al., 2022; Bui Thi Mai & Lampert, 2021; Boursier et al., 2022; Kou et al., 2023), as it provides essential simplifications that facilitate deriving theoretical insights. We further support this assumption with numerical evidence, presented in Appendix I.1. Finally, *Assumption 2.4* is motivated by recent theoretical results on the implicit bias of gradient descent in multiclass classification (Ravi et al., 2024; Lyu et al., 2021), as discussed in §2.1.

**Implications.** The assumptions introduced above lead to a clean closed-form expression for the $\bar{K}$-class hard-margin SVM solution for the fine-tuning task, and also imply orthogonality between the pre-trained weights $W_{\mathrm{pre}}$ and the fine-tuning dataset $\mathcal{D}_{\mathrm{ft}}$, which lays a foundation for our subsequent theoretical analysis in §3. To formalize these implications, we begin by introducing the concept of a *simplex equiangular tight frame* (simplex ETF), which characterizes the geometry of the hard-margin SVM solution in our setting.

**Definition 2.1** (Simplex ETF). *A $m$-simplex ETF is a collection of $m$ vectors in $\mathbb{R}^d$ given by the columns of $M_m := \sqrt{\frac{m}{m-1}} P \left( \boldsymbol{I}_m - \frac{1}{m} \boldsymbol{1}_m \boldsymbol{1}_m^\top \right)$, where $P \in \mathbb{R}^{d \times m}$ satisfies $P^\top P = \boldsymbol{I}_m$.*

Using this definition, we now present a proposition that characterizes the structure of the hard-margin SVM solution for the fine-tuning data ($\mathcal{D}_{\mathrm{ft}}$), and the orthogonality between the pre-trained weight ($W_{\mathrm{pre}}$) and fine-tuning data, under Assumptions 2.3 and 2.4.

**Proposition 2.1.** *Under Assumptions 2.3 and 2.4, the following properties hold:*

(i) *Under condition (a) of Assumption 2.3, the $\bar{K}$-class hard-margin SVM solution for the fine-tuning task $\mathcal{D}_{\mathrm{ft}}$, defined as*

$$\bar{W}_{\mathrm{ft}}^{\mathrm{SVM}} = \arg\min_{W \in \mathbb{R}^{\bar{K} \times d}} \frac{1}{2}\|W\|_F^2 \quad \textit{subject to} \quad C(W; \mathcal{D}_{\mathrm{ft}}) \geq 1,$$

*admits the closed-form solution: $\bar{W}_{\mathrm{ft}}^{\mathrm{SVM}} = \left( M_{\bar{K}} \otimes \boldsymbol{1}_n^\top \right) X_{\mathrm{ft}}^\top$, where $\boldsymbol{x}_{i,j}$ denoting the $j$-th sample from class $i$, and $X_{\mathrm{ft}} = [\boldsymbol{x}_{1,1}, \ldots, \boldsymbol{x}_{1,n}, \ldots, \boldsymbol{x}_{\bar{K},n}]$ is the fine-tuning data matrix.*

(ii) *Under condition (b) of Assumption 2.3 and Assumption 2.4, the pre-trained weights are orthogonal to the fine-tuning data: $W_{\mathrm{pre}}\bar{\boldsymbol{x}} = 0, \quad \forall \bar{\boldsymbol{x}} \in \mathcal{D}_{\mathrm{ft}}$.*

**Remark on Proposition 2.1.** The first result in Proposition 2.1 reveals that the hard-margin SVM solution for the fine-tuning task admits a compact closed-form expression: $W_{\mathrm{ft}}^{\mathrm{SVM}} = \left(M_{\bar{K}} \otimes \mathbf{1}_n^\top\right) X_{\mathrm{ft}}^\top$. However, this form does not immediately reveal how the classifier aggregates information from the fine-tuning data. To provide insight, we explicitly derive the expression for the *first row* of $W_{\mathrm{ft}}^{\mathrm{SVM}}$ (the remaining rows follow symmetrically), yielding:

$$\sqrt{\tfrac{\bar{K}}{\bar{K}-1}} \, P \left( \tfrac{\bar{K}-1}{\bar{K}} \sum_{i=1}^n \bar{\boldsymbol{x}}_{1,i} - \tfrac{1}{\bar{K}} \sum_{k=2}^{\bar{K}} \sum_{j=1}^n \bar{\boldsymbol{x}}_{k,j} \right).$$

Intuitively, each row of $W_{\mathrm{ft}}^{\mathrm{SVM}}$ encodes a direction that emphasizes its corresponding class mean while uniformly suppressing the influence of all other classes, leading to class separation in max-margin classification. The second result in Proposition 2.1 arises naturally due to the representer theorem for SVMs, which implies that the hard-margin SVM solution for pre-training task is a linear combination of pre-training data only. Under condition (b) in Assumption 2.3, each row of $W_{\mathrm{pre}}$ is consequently orthogonal to the fine-tuning data.

Building upon these assumptions and propositions, we develop a rigorous theoretical framework that for the first time provides a principled explanation for LoRA's empirical advantages in the subsequent section. Our analysis reveals properties that enables precise quantification of LoRA's benefits in terms of margin preservation and multi-task performance.

## 3 MAIN RESULTS

In this section, we first analyze the optimization landscape of the LoRA objective and characterize its global minimum in §3.1. We then derive closed-form expressions for the margins on the union of pre-training and fine-tuning datasets, demonstrating why LoRA mitigates *catastrophic forgetting* in §3.2. Finally, in §3.3, we extend our margin-based analysis to the multi-task setting, providing the first theoretical explanation for the effectiveness of *adapter merging* and deriving optimal mixing coefficients that maximize the margin on the union of all tasks.

### 3.1 GLOBAL MINIMUM OF LORA OBJECTIVE

In this section, we present our main theoretical result on the characterization of the global minimizer of the LoRA objective in (6). Interestingly, under suitable regularization, we find that part of the optimal LoRA adapters aligns exactly with the $\bar{K}$-class hard-margin SVM solution ($\bar{W}_{\mathrm{ft}}^{\mathrm{SVM}}$) for the fine-tuning data (see Proposition 2.1 for definition). While our analysis does not assume or rely on this, its emergence highlights a geometric alignment that partially explains LoRA's effectiveness.

**Theorem 3.1.** *There exists a critical regularization weight* $\lambda_{\mathrm{crit}} \in (0, \frac{1}{K\sqrt{n}})$ *and scalar functions* $a_\lambda, b_\lambda, c_\lambda, \Theta_\lambda$ *of* $\lambda$ *such that for any global minimizer* $\left(B_\lambda^*, A_\lambda^*\right)$ *of* (6) *with* $\lambda > 0$, *we have*

$$B_\lambda^* A_\lambda^* = \left( \begin{pmatrix} (a_\lambda + b_\lambda)\, \mathbf{I}_{\bar{K}} \; - \; b_\lambda \, \mathbf{1}_{\bar{K}} \mathbf{1}_{\bar{K}}^\top \\ -c_\lambda \, \mathbf{1}_{K-\bar{K}} \mathbf{1}_{\bar{K}}^\top \end{pmatrix} \otimes \mathbf{1}_n^\top \right) X_{\mathrm{ft}}^\top, \quad \|B_\lambda^* A_\lambda^*\|_F = \Theta_\lambda. \tag{7}$$

*When* $\bar{K} \geq 2$, *the scalar functions* $a_\lambda, b_\lambda, c_\lambda, \Theta_\lambda$ *are characterized as follows:*

(i) *High-penalty regime* $\left(\lambda \geq \frac{1}{K\sqrt{n}}\right)$: $a_\lambda = b_\lambda = c_\lambda = \Theta_\lambda = 0$, *thus* $B_\lambda^* A_\lambda^* = \mathbf{0}_{K \times d}$.

(ii) *Intermediate regime* $\left(\lambda_{\mathrm{crit}} < \lambda < \frac{1}{K\sqrt{n}}\right)$: $a_\lambda = \frac{\Theta_\lambda}{\sqrt{\bar{K}n}}, b_\lambda = \frac{\Theta_\lambda}{\bar{K}\sqrt{\bar{K}n}}, c_\lambda = 0$, *and* $\Theta_\lambda$ *is the unique root of a nonlinear equation (see* (28) *in Appendix E), thus the minimizer is*

$$B_\lambda^* A_\lambda^* = \frac{\Theta_\lambda}{\sqrt{\bar{K}n}} \left( \begin{pmatrix} M_{\bar{K}} \\ \mathbf{0}_{(K-\bar{K}) \times \bar{K}} \end{pmatrix} \otimes \mathbf{1}_n^\top \right) X_{\mathrm{ft}}^\top. \tag{8}$$

(iii) *Low-penalty regime* $\left(\lambda \leq \lambda_{\mathrm{crit}}\right)$: *in general* $a_\lambda, b_\lambda, c_\lambda, \Theta_\lambda$ *are positive. Nevertheless,*

$$\lim_{\lambda \to 0^+} \Theta_\lambda = \infty, \qquad \lim_{\lambda \to 0^+} \frac{B_\lambda^* A_\lambda^*}{\|B_\lambda^* A_\lambda^*\|_F} = \sqrt{\frac{1}{nK}} \left( M_K^{(\bar{K})} \otimes \mathbf{1}_n^\top \right) X_{\mathrm{ft}}^\top, \tag{9}$$

*where* $K$ *is the number of pre-training classes and* $M_K^{(\bar{K})}$ *is the first* $\bar{K}$ *columns of the K-ETF matrix* $M_K$.

The proof of the above theorem is provided in Appendix E. Theorem 3.1 shows that the product of the optimal LoRA adapter has a unified form as is shown in (7), and identifies three different regimes based on the regularization parameter $\lambda$. We make the following remarks:

**Effect of regularization and connection with max-margin classifier.** In the above theorem, the regularization parameter determines the structure of the optimal LoRA adapters. In the *high-penalty regime* (large regularization), the regularization term dominates, forcing optimal adapters toward zero. In the *intermediate regime* (medium regularization), the LoRA adapters balance between minimizing cross-entropy and regularization, resulting in the explicit structure given in (8). Notably, the first $\bar{K}$ rows of the product $B_\lambda^* A_\lambda^*$ align with the $\bar{K}$-class hard-margin SVM solution for the fine-tuning data (see Proposition 2.1 for comparison). This alignment offers a clear interpretation of the learned solution and partially explains the strong empirical performance of LoRA in practice. In the *low-penalty regime* (small regularization), explicit solutions for the scalar functions $a_\lambda, b_\lambda$ are difficult to derive. However, as $\lambda \to 0^+$, the cross-entropy term dominates, pushing the optimal solution toward infinity, as indicated by $\lim_{\lambda \to 0^+} \Theta_\lambda = \infty$. Additionally, our asymptotic analysis characterizes the limiting direction of $B_\lambda^* A_\lambda^*$ as is shown in (9). We emphasize that this asymptotic direction does not align with the hard-margin SVM solution for the fine-tuning data unless $\bar{K} = K$.

**Implication for fine-tuned model.** The structured form of the optimal LoRA adapters in (7) naturally enables $W_{\text{LoRA}}^\lambda := W_{\text{pre}} + B_\lambda^* A_\lambda^*$ to perform well on both pre-training and fine-tuning data. Under Assumptions 2.3 and 2.4, Proposition 2.1 and Theorem 3.1 imply that $W_{\text{pre}} \bar{\boldsymbol{x}} = 0$ for all $\bar{\boldsymbol{x}} \in \mathcal{D}_{\text{ft}}$ and $B_\lambda^* A_\lambda^* \boldsymbol{x} = 0$ for all $\boldsymbol{x} \in \mathcal{D}_{\text{pre}}$. That is, the two components operate independently: $W_{\text{pre}}$ classifies the pre-training data, while $B_\lambda^* A_\lambda^*$ adapts to the fine-tuning data:

$$\forall \boldsymbol{x} \in \mathcal{D}_{\text{pre}} \cup \mathcal{D}_{\text{ft}}, \quad W_{\text{LoRA}}^\lambda \boldsymbol{x} = \begin{cases} W_{\text{pre}} \boldsymbol{x} & \text{if } \boldsymbol{x} \in \mathcal{D}_{\text{pre}}, \\ B_\lambda^* A_\lambda^* \boldsymbol{x} & \text{if } \boldsymbol{x} \in \mathcal{D}_{\text{ft}}. \end{cases} \quad \text{holds} \quad \forall \lambda > 0. \quad (10)$$

This clean separation allows us to derive the margin of $W_{\text{LoRA}}^\lambda$ for any data in the combined dataset:

$$\begin{cases} \frac{W_{\text{pre}} \boldsymbol{x}}{\|W_{\text{pre}}\|_F} \cdot \frac{\|W_{\text{pre}}\|_F}{\|W_{\text{LoRA}}^\lambda\|_F} & \text{if } \boldsymbol{x} \in \mathcal{D}_{\text{pre}}, \\ \frac{B_\lambda^* A_\lambda^* \boldsymbol{x}}{\|B_\lambda^* A_\lambda^* \boldsymbol{x}\|_F} \cdot \frac{\|B_\lambda^* A_\lambda^* \boldsymbol{x}\|_F}{\|W_{\text{LoRA}}^\lambda\|_F} & \text{if } \boldsymbol{x} \in \mathcal{D}_{\text{ft}}. \end{cases} \quad (11)$$

The expression in (11) shows that the normalized margin on the combined dataset depends not only on the individual normalized margins of $W_{\text{pre}}$ and $B_\lambda^* A_\lambda^*$ on the $\mathcal{D}_{\text{pre}}$ and $\mathcal{D}_{\text{ft}}$ respectively, but also on the relative Frobenius norms of these components. In the following sections, we use (11) to derive closed-form expressions for the normalized margin on the combined dataset and to compute the optimal mixing coefficients for *adapter merging*. For clarity of presentation, we focus on the case $\bar{K} = K$ and refer the reader to Appendix G and Appendix H for the full version of our results.

### 3.2 LoRA PROVABLY REDUCES FORGETTING

In this section, we demonstrate that LoRA provably reduces forgetting through the lens of normalized margin, and identify an optimal regularization level achieving max-margin over the combined datasets. For convenience, we define the following shorthand for margins: $\gamma_{\text{pre}} = \gamma(W_{\text{pre}}; \mathcal{D}_{\text{pre}}), \gamma_{\text{ft},\lambda} = \gamma(B_\lambda^* A_\lambda^*; \mathcal{D}_{\text{ft}})$ (motivated by (11)).

With these definitions in place, we now present our main theorem.

**Theorem 3.2.** *Adopt the setup of Theorem 3.1, let $\gamma_{\text{ft}}^*$ be the max normalized margin any linear classifier can obtain on the fine-tuning data $\mathcal{D}_{\text{ft}}$, and recall the scalar $\rho_{\text{pre}}$ from Assumption 2.4. Then, the normalized margins of $W_{\text{LoRA}}^\lambda$ on the union of pre-training and fine-tuning data can be characterized uniformly over all $\lambda$ as follows:*

$$\gamma(W_{\text{LoRA}}^\lambda; \mathcal{D}_{\text{pre}}) = \gamma_{\text{pre}} \frac{\rho_{\text{pre}}}{\sqrt{\Theta_\lambda^2 + \rho_{\text{pre}}^2}}, \quad \gamma(W_{\text{LoRA}}^\lambda; \mathcal{D}_{\text{ft}}) = \gamma_{\text{ft},\lambda} \frac{\Theta_\lambda}{\sqrt{\Theta_\lambda^2 + \rho_{\text{pre}}^2}},$$

$$\gamma(W_{\text{LoRA}}^\lambda; \mathcal{D}_{\text{pre}} \cup \mathcal{D}_{\text{ft}}) = \min \left\{ \gamma(W_{\text{LoRA}}^\lambda; \mathcal{D}_{\text{pre}}), \gamma(W_{\text{LoRA}}^\lambda; \mathcal{D}_{\text{ft}}) \right\} \quad (12)$$

*Moreover, $\Theta_\lambda, \gamma_{\text{ft},\lambda}$ take different values depending on the regime $\lambda$ is in:*

(i) *High–penalty regime:* $\lambda \geq \frac{1}{K\sqrt{n}}$: $\Theta_\lambda = \gamma_{\text{ft},\lambda} = 0$.

(ii) *Intermediate and low-penalty regime:* $\lambda < \frac{1}{K\sqrt{n}}$: $\gamma_{\text{ft},\lambda} = \gamma_{\text{ft}}^*$, *and* $\Theta_\lambda$ *is a decreasing function.*

**Optimal trade-off choice of** $\lambda$**.** *There exists a unique* $\lambda^*$ *such that*

$$\max_{\lambda>0} \; \gamma(W_{\text{LoRA}}^\lambda; \mathcal{D}_{\text{pre}} \cup \mathcal{D}_{\text{ft}}) \; = \; \left( \frac{1}{(\gamma_{\text{ft}}^*)^2} + \frac{1}{\gamma_{\text{pre}}^2} \right)^{-1/2}, \qquad \text{attained at } \lambda = \lambda^*. \qquad (13)$$

**Remark 3.1.** *Notably, the intermediate and low-penalty regimes coincide when* $\bar{K} = K$. *This merging is specific to the case* $\bar{K} = K$; *when* $\bar{K} < K$, *the two regimes remain distinct.*

**Interpretation of the normalized margin.** Theorem 3.2 characterizes the margins of the LoRA fine-tuned model $W_{\text{LoRA}}^\lambda$ on the pre-training and fine-tuning datasets. Notably, the margin on the pre-training data remains positive in both the *intermediate* and *low-penalty* regimes, indicating that LoRA mitigates *catastrophic forgetting*. Furthermore, the margin on the combined dataset, given in (12), is defined as the minimum of two terms: the margin on the pre-training data weighted by the relative magnitude of the pre-trained weights, and the margin on the fine-tuning data weighted by the norm of the LoRA adapters. This structure reveals a clear trade-off: decreasing the regularization parameter $\lambda$ increases the adapter norm $\Theta_\lambda$, which decreases the pre-training margin contribution while increasing the fine-tuning margin contribution.

**Uniqueness of the optimal** $\lambda$**.** Due to the opposing effects of $\Theta_\lambda$ on the two components of the combined margin, the maximum of $\gamma(W_{\text{LoRA}}^\lambda; \mathcal{D}_{\text{pre}} \cup \mathcal{D}_{\text{ft}})$ is achieved when the weighted margins are equal. This balance determines a unique value of $\Theta_\lambda$. Since $\Theta_\lambda$ is a strictly decreasing function, it follows that there exists a unique $\lambda$ that maximizes the normalized margin on the combined datasets.

**Full fine-tuning as an alternative method.** While our work focuses on LoRA, it is instructive to briefly contrast it with the more conventional strategy of full fine-tuning. Full fine-tuning updates all parameters of the pretrained model, which is considerably less efficient in both computation and storage, whereas LoRA achieves adaptation through a compact low-rank parameterization. In the linear classification setting, the two approaches correspond to the following regularized objectives:

$$\min_{B,A} \; L(W_{\text{pre}} + BA; \mathcal{D}_{\text{ft}}) + \frac{\lambda}{2}\left(\|B\|_F^2 + \|A\|_F^2\right), \qquad \text{LoRA objective}, \qquad (14)$$

$$\min_{W} \; L(W; \mathcal{D}_{\text{ft}}) + \frac{\lambda}{2}\|W - W_{\text{pre}}\|_F^2, \qquad \text{Full fine-tuning objective}. \qquad (15)$$

Beyond efficiency, the two formulations induce fundamentally different (implicit) biases. By exploiting the variational form of the nuclear norm (Recht et al., 2010), one can show that for any solution $(B^*, A^*)$ of (14), the induced update $W_{\text{LoRA}}^\lambda$ is equivalently the solution of

$$\min_{W,\text{rank}(W-W_{\text{pre}})\leq r} L(W; \mathcal{D}_{\text{ft}}) + \frac{\lambda}{2}\|W - W_{\text{pre}}\|_*, \qquad \text{Implicit bias of LoRA fine-tuning}, \quad (16)$$

which highlights LoRA's connection to nuclear-norm regularization under rank constraints. We refer the readers to Jang et al. (2024) for the proof of the argument. Understanding the consequences of this distinction of nuclear norm regularization and Frobenius norm regularization remains an open direction for future work. Importantly, our goal is not to compare LoRA and full fine-tuning, but to rigorously analyze why LoRA exhibits reduced forgetting and to provide principled guidance for selecting regularization parameters in adapter-based fine-tuning.

### 3.3 LoRA supports adapter merging

In this section, we explain why LoRA supports *adapter merging* through the same margin lens.

We now consider a scenario where a single pre-trained network is fine-tuned independently on $T$ datasets, denoted by $\mathcal{D}_1, \ldots, \mathcal{D}_T$. Let $(B_i^*, A_i^*)$ represent the LoRA adapter obtained from training on dataset $\mathcal{D}_i$, for each $i \in [T]$. Our goal is to merge these adapters into a unified model that performs well simultaneously on the original pre-training task and on all fine-tuning tasks $\mathcal{D}_1, \ldots, \mathcal{D}_T$. To achieve this, we merge all adapters with the pre-trained weight matrix as follows:

$$W_{\text{LoRA}}(\boldsymbol{\alpha}) := W_{\text{pre}} + \sum_{i=1}^T \alpha_i B_i^* A_i^*, \qquad \alpha_1, \ldots, \alpha_T \in \mathbb{R}, \qquad (17)$$

where $\boldsymbol{\alpha} = (\alpha_1, \ldots, \alpha_T)$ are user-specified mixing coefficients. Moreover, we generalize the Assumption 2.3 to the setting of learning multiple tasks as follows.

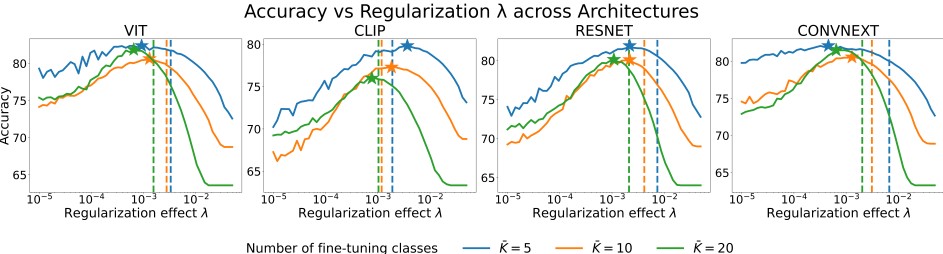

Figure 1: Accuracy of the fine-tuned model across varying regularization levels for four different pre-trained models. Each line represents a fine-tuning task, indicated by color. Stars highlight the best-performing regularization value for each task. Vertical dashed lines indicate the theoretically predicted optimal regularization parameter for each setting. Alignment between the star and the dashed line of the same color reflects how well our theory predicts the empirically optimal regularization level.

**Assumption 3.1.** *For each fine-tuning task $\mathcal{D}_i$ the dataset consists of $m_i$ classes, each containing $n_i$ samples. All fine-tuning feature vectors have unit norm, are pairwise orthogonal, and are orthogonal to every pre-training feature vector.*

The following theorem characterizes the optimal mixing coefficients. See Appendix H for the proof.

**Theorem 3.3.** *Under Assumption 3.1, and suppose each fine–tuning task $\mathcal{D}_i$ $(i = 1, \ldots, T)$ is trained with a regularization parameter in the intermediate or low-penalty regime. Let $\Theta_{\lambda,i} = \|B_i^* A_i^*\|_F$ and $\gamma_i = \gamma(B_i^* A_i^*; \mathcal{D}_i)$. For an arbitrary coefficient vector $\boldsymbol{\alpha} = (\alpha_1, \ldots, \alpha_T)$ the merged model* (17) *achieves the normalized margins on each task as follows*

$$\gamma\big(W_{\mathrm{LoRA}}(\boldsymbol{\alpha}); \mathcal{D}_{\mathrm{pre}}\big) = \frac{\gamma_{\mathrm{pre}}\rho_{\mathrm{pre}}}{\sqrt{\rho_{\mathrm{pre}}^2 + \sum_{j=1}^T \alpha_j^2 \Theta_{\lambda,i}^2}}, \quad \gamma\big(W_{\mathrm{LoRA}}(\boldsymbol{\alpha}); \mathcal{D}_i\big) = \frac{\gamma_i \alpha_i \Theta_{\lambda,i}}{\sqrt{\rho_{\mathrm{pre}}^2 + \sum_{j=1}^T \alpha_j^2 \Theta_{\lambda,i}^2}},$$

$$\gamma\big(W_{\mathrm{LoRA}}(\boldsymbol{\alpha}); \mathcal{D}_{\mathrm{pre}} \cup \{D_i\}_{i=1}^T\big) = \min\big\{\gamma\big(W_{\mathrm{LoRA}}(\boldsymbol{\alpha}); \mathcal{D}_{\mathrm{pre}}\big), \gamma\big(W_{\mathrm{LoRA}}(\boldsymbol{\alpha}); \mathcal{D}_i\big)\big\}, i \in [T]$$

*Choosing the weights $\alpha_i = \frac{\rho_{\mathrm{pre}}\gamma_{\mathrm{pre}}}{\gamma_i \Theta_{\lambda,i}}$, $i \in [T]$, maximizes the margin on the union of all tasks:*

$$\max_{\boldsymbol{\alpha}} \gamma\big(W_{\mathrm{LoRA}}(\boldsymbol{\alpha}); \mathcal{D}_{\mathrm{pre}} \cup \{D_i\}_{i=1}^T\big) = \left(\frac{1}{\gamma_{\mathrm{pre}}^2} + \sum_{j=1}^T \frac{1}{(\gamma_j)^2}\right)^{-1/2}.$$

Theorem 3.3 characterizes the optimal mixing coefficients, i.e., $\frac{\rho_{\mathrm{pre}}\gamma_{\mathrm{pre}}}{\gamma_i \Theta_{\lambda,i}}$, based on two key quantities: the ratio between the Frobenius norms of the pre-trained weight matrix and each LoRA adapter product, i.e., $\rho_{\mathrm{pre}}/\Theta_{\lambda,i}$, and the ratio between the margins for the pre-training task and for each task, i.e., $\gamma_{\mathrm{pre}}/\gamma_i$. As either ratio decreases, the optimal mixing coefficient for the corresponding adapter should increase. Intuitively speaking, a larger $\rho_{\mathrm{pre}}/\Theta_{\lambda,i}$ implies the adapter has a weaker impact relative to the pre-trained model, necessitating a larger weighting to achieve a balanced contribution. Similarly, a larger $\gamma_{\mathrm{pre}}/\gamma_i$ indicates that task $i$ is inherently more challenging compared to the pre-trained task, thus requiring a larger weight for its adapter to ensure satisfactory performance in the unified model, i.e., $W_{\mathrm{LoRA}}(\boldsymbol{\alpha})$. Finally, we point out that the optimal mixing coefficients proposed in Theorem 3.3 can be computed after the training as long as the regularization parameter lies in the *intermediate regime*, making the approach practical and easy to implement.

So far, we have characterized the optimal solutions to the LoRA objective in §3.1, explained why LoRA mitigates *catastrophic forgetting* by analyzing margins on pre-training and fine-tuning datasets in §3.2, and derived optimal mixing coefficients for adapter merging through the same margin perspective in §4.2. In the next section, we numerically validate these theoretical insights.

## 4 EXPERIMENTS

In this section, we evaluate our theoretical predictions in realistic settings where assumptions are not strictly satisfied (Assumption 2.3 and Assumption 2.4). Specifically, we test: *(i)* the effect of regularization on LoRA's performance across architectures and tasks, and *(ii)* the accuracy of our predicted mixing coefficients for adapter merging.

**Setup.** We use four popular pre-trained models, `ResNet-50`, `ViT-B/16`, `ConvNeXt`, and `CLIP` (He et al., 2016; Dosovitskiy et al., 2021; Liu et al., 2022; Radford et al., 2021), as frozen

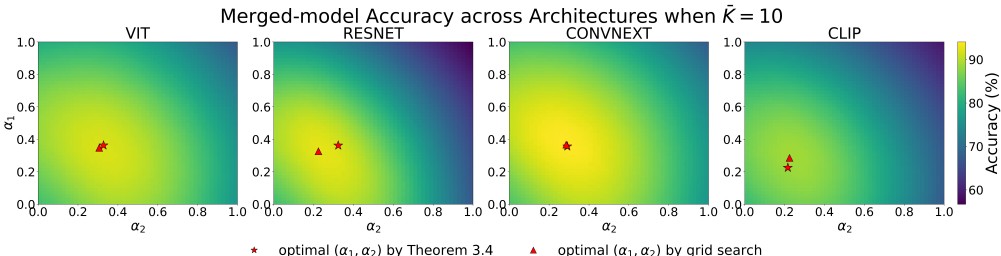

Figure 2: Merged-model accuracy across architectures. Each panel shows the accuracy of the merged model evaluated on the combined dataset, across a $50 \times 50$ grid of mixing coefficients $(\alpha_1, \alpha_2) \in (0, 1)^2$. For each architecture, the red star indicates the theoretically predicted optimal coefficients from Theorem 3.3, while the red triangle marks the empirically optimal coefficients.

feature extractors on CIFAR-100 (Krizhevsky et al., 2009). We apply LoRA to the final linear classification layer only, following the standard linear probing protocol (Kornblith et al., 2019). Tasks are constructed using CIFAR-100's superclass hierarchy. All training and task construction details are provided in Appendix I.

**Metric.** Though our theory is developed for normalized margin, real-world classification problems are more challenging, and perfect accuracy is typically unattainable, which leads to negative normalized margin of the fine-tuned model. Therefore, we report classification accuracy as a more practical measure of performance in this setting.

### 4.1 EFFECT OF REGULARIZATION

We first pre-train a linear classifier on the pretraining task, and then fine-tune it using LoRA across 50 logarithmically spaced regularization strengths, with $\lambda \in [10^{-5}, 10^{-1}]$. Figure 1 shows that LoRA's performance exhibits a non-monotonic relationship with $\lambda$, peaking at a moderate value. Notably, our theoretically predicted optimal $\lambda$ aligns well with the best empirical choice when the pre-trained models are VIT, RESNET and CLIP. Detailed comparisons are reported in Appendix I.

### 4.2 OPTIMAL MIXING COEFFICIENTS FOR ADAPTER MERGING: THEORY VS. GRID SEARCH

We fine-tune LoRA adapters on two disjoint tasks $(\mathcal{D}_1, \mathcal{D}_2)$, then merge them with the pre-trained weights using mixing coefficients $(\alpha_1, \alpha_2)$. We evaluate the merged model's performance on the combined dataset $\mathcal{D}_{\text{pre}} \cup \mathcal{D}_1 \cup \mathcal{D}_2$. As shown in Figure 2, our predicted mixing coefficients closely match the grid-searched optima, confirming the practical accuracy of our merging theory. In Appendix I, we provide more experimental results when $\bar{K} = 5, 20$.

## 5 CONCLUSION, LIMITATIONS, AND FUTURE WORK

In this work, we provide a theoretical analysis of LoRA fine-tuning through the lens of normalized margin. We characterize the structure of optimal LoRA adapters under varying regularization regimes and show how LoRA mitigates catastrophic forgetting. Our results reveal a clean separation of responsibility between the pre-trained weights and LoRA adapters, enabling margin-based analysis on pre-training and fine-tuning tasks. We further extend our framework to the setting of adapter merging and derive closed-form expressions for the optimal mixing coefficients. Empirical results across multiple architectures validate our theoretical predictions, even when the underlying assumptions (e.g., data orthogonality) are mildly violated in practice.

**Limitations and future work.** Our analysis relies on several simplifying assumptions. In particular, we assume orthogonal task structure (*Assumption 2.3*), perfect alignment between pre-trained weights and pre-training tasks (*Assumption 2.4*), and balanced class sizes in the fine-tuning data. While these assumptions enable closed-form characterization, they limit the generality of our results. An important direction for future work is to relax these constraints and explore how LoRA behaves under more realistic data distributions and pre-training conditions. Extending the margin-based perspective to non-linear models is another promising avenue. Finally, as discussed in §3.2, full fine-tuning and LoRA fine-tuning induce different implicit biases. An important open direction is to characterize the regimes in which LoRA has an advantage over full fine-tuning, for example by comparing their respective generalization errors.

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

## A    USAGE OF LARGE LANGUAGE MODEL

We used GPT-5 to assist with revising the writing and setting up the basic experimental pipeline (e.g., loading pretrained models and extracting features). All algorithmic implementations were written by us.

## B    RELATED WORK

**Theory of LoRA.** There are many works theoretically studying the expressiveness, characterizing the loss landscape, and understanding the learning dynamics of LoRA. Zeng & Lee (2023) prove that, under mild assumptions, LoRA can approximate any deep linear, feed-forward, or transformer network. Within the NTK regime, Malladi et al. (2023) characterize the conditions under which one can study LoRA in the NTK regime, while Jang et al. (2024) show that when the LoRA rank is $r_i \gtrsim \sqrt{N}$, where $N$ is the number of samples, the optimization landscape of LoRA has no spurious local minima, and GD can find $\mathcal{O}(\sqrt{N})$-rank solutions that generalize well. Moreover, Xu et al. (2025) study the learning dynamics of LoRA in the context of matrix factorization, and show that smaller initialization leads to longer training time and lower training error. However, none of these studies theoretically characterize why LoRA reduces *catastrophic forgetting* and support effective *adapter merging*.

**LoRA merging.** A variety of techniques have been proposed to *merge* task-specific LoRA adapters by forming a weighted average of their parameters. For example, *LoRA-Hub* (Huang et al., 2023) learns per-task mixing coefficients first, and then applies a weighted average of LoRA weight matrices $B$ and $A$ separately. *LoRA-Flow* (Wang et al., 2024) introduces token-level gates that dynamically assign merging coefficients to each LoRA adapter before taking weighted averaging. *LoRA-Retriever* (Zhao et al., 2024) first retrieves the most relevant adapters for each input, and then averages the selected LoRA adapters. Beyond weighted averaging, another line of work attempts to address redundancy and conflicting updates directly. *TIES* (Yadav et al., 2023) proceeds in three steps: trimming redundant parameters, resolving sign conflicts into an aggregate vector, and averaging only the parameters consistent with the aggregate sign, thereby mitigating degradation from redundant or conflicting updates. *DARE* (Yu et al., 2024) can be used as a preprocessing step for other merging methods, where parameters are randomly dropped according to a specified rate and the remaining ones are rescaled, reducing redundancy and potential interference among merged adapters. However, all these methods remain largely heuristic and lack theoretical guarantees.

**Regularization and task orthogonality help reduce forgetting.** A large body of work in fine-tuning and continual learning shows that *controlling deviation of the fine-tuned model from the pretrained model* reduces catastrophic forgetting. Concretely, these approaches impose *weight-space* penalties that restrict the magnitude and direction of parameter updates away from the pretrained solution. Weight-anchoring and importance-aware penalties exemplify this idea: Kirkpatrick et al. (2017) penalize movement along directions identified by the Fisher information matrix, while Zenke et al. (2017) accumulate path-wise importance during training. Schwarz et al. (2018) extend this idea to long sequences of tasks. Related approaches estimate parameter sensitivity (Aljundi et al., 2018), apply output-level distillation to preserve prior behavior (Li & Hoiem, 2016), or explicitly regularize parameters toward the pretrained anchor (Li et al., 2018). Although these approaches differ in how importance or anchoring is computed, the common mechanism is the same: *weight-space regularizers constrain updates during fine-tuning relative to the pretrained model to mitigate catastrophic forgetting*. Another complementary perspective highlights the role of task relations. When task representations are sufficiently decorrelated, interference between tasks is naturally reduced and forgetting is alleviated. In linear and kernelized models, it has been shown that when tasks are orthogonal or nearly orthogonal, cross-task gradient interference vanishes in expectation, implying negligible forgetting (Doan et al., 2021; Evron et al., 2022). These insights resonate with our analysis of LoRA, where we explicitly study a regularized objective on low-rank adapters and make the simplifying assumption that the fine-tuning data are orthogonal to the pretraining data. This assumption directly connects to prior findings on task orthogonality and provides a tractable setting in which we can isolate and analyze the role of regularization in mitigating forgetting.

## C PRELIMINARY LEMMAS

In this section, we provide several preliminary lemmas that will be used in the proof.

**Lemma C.1** (Theorem 3 in Riedel (1992)). *Let $A \in \mathbb{R}^{l \times l}$ with $\mathrm{rank}(A) < l$, and $v_i, w_i, i = 1, 2$ be vectos in $\mathbb{R}^l$. Let $v_1 \in M(A), w_1 \perp M(A)$, and $v_2 \in M(A^*), w_2 \perp M(A^*)$, where $M(A)$ denotes the range of $A$. Assume $w_2 \parallel w_1$ and $w_i \neq 0, i = 1, 2$. Let $\Omega = A + (v_1 + w_1)(v_2 + w_2)^*$, Then*

$$\Omega^\dagger = A^\dagger - \frac{w_2 v_2^* A^\dagger}{\|w_2\|^2} - \frac{A^\dagger v_1 w_1^*}{\|w_1\|^2} + (1 + v_2^* A^\dagger v_1) \cdot \frac{w_2 w_1^*}{\|w_1\|^2 \|w_2\|^2}\,.$$

We refer the readers to Riedel (1992) for the proof.

**Lemma C.2** (Variational form of the nuclear norm). *For any fixed $Z \in \mathbb{R}^{K \times m}$, we have*

$$\|Z\|_* = \min_{Z = BA} \frac{1}{2}(\|B\|_F^2 + \|A\|_F^2)\,.$$

We refer the readers to Recht et al. (2010) for the proof.

**Lemma C.3.** *For any fixed $Z \in \mathbb{R}^{K \times m}, K \geq m$, we have*

$$\|Z\|_* = \max_{V \in \mathbb{R}^{m \times m}, V^\top V = I_m} \mathrm{Tr}(ZV)\,,$$

*if $K < m$, we have*

$$\|Z\|_* = \max_{U \in \mathbb{R}^{K \times K}, U^\top U = I_K} \mathrm{Tr}(UZ)\,.$$

*Moreover, for any two matrices $A, B \in \mathbb{R}^{K \times m}$, one has*

$$\sum_{i=1}^m \sigma_i(AB^\top) \leq \sum_{i=1}^m \sigma_i(A)\sigma_i(B)\,.$$

We refer the readers to Horn & Johnson (2012) for the proof.

**Lemma C.4** (Berge's Maximum Theorem). *Let $X, \Theta$ be topological spaces, $f : X \times \Theta \to \mathbb{R}$ be a continuous function on the product $X \times \Theta$, and $C : \Theta \to X$ be a compact-valued correspondence such that $C(\theta) \neq \emptyset$ for all $\theta \in \Theta$. Define the marginal function $f^* : \Theta \to \mathbb{R}$ and the set of minimizers $C^* : \Theta \to X$ by*

$$f^*(\theta) = \sup\{f(x, \theta) : x \in C(\theta)\}$$
$$C^*(\theta) = \arg\max\{f(x, \theta) : x \in C(\theta)\} = \{x \in C(\theta) : f(x; \theta) = f^*(\theta)\}\,.$$

*If $C$ is continuous at $\theta$, then the value function $f^*(\theta)$ is continuous, and the set of maximizers $C^*(\theta)$ is upper-hemicontinuous with nonempty and compact values. Moreover, if $C^*(\theta)$ is single-valued, and thus is a continuous function rather than a correspondence.*

We refer the readers to Sundaram (1996) for the proof.

**Lemma C.5** (Invariance under Permutation). *Suppose $X$ satisfies $\Pi X \Pi^\top = X$ for any permutation $\Pi$, then $X = a\boldsymbol{I} + c\boldsymbol{1}\boldsymbol{1}^\top$ for any constants $a, c \in \mathbb{R}$.*

We refer the readers to Lemma 10 in Hong & Ling (2023) for the proof.

**Proposition C.1** (Representer Theorem for Hard-margin SVM Problem). *Let $\{(\boldsymbol{x}_{c,i}, \boldsymbol{y}_c)\}_{i=1,c=1}^{n_c, K}$ be the dataset with features $\boldsymbol{x}_{c,i} \in \mathbb{R}^d$ and class labels $\boldsymbol{y}_c \in \mathbb{R}^K$. Then, let $W^{\mathrm{mm}}$ be the solution of the following hard-margin SVM problem, i.e.,*

$$\min_{W \in \mathbb{R}^{K \times d}} \frac{1}{2}\|W\|_F^2 \quad s.t. \quad \forall i \in [K],\ \forall j \in [n_i],\ \forall k \neq i:\ \boldsymbol{w}_i^\top \boldsymbol{x}_{i,j} \geq \boldsymbol{w}_k^\top \boldsymbol{x}_{i,j} + 1\,,$$

*where $W = [\boldsymbol{w}_1, \cdots, \boldsymbol{w}_K]^\top$. Then, there exists a set of scalars $\{\alpha_{i,c,j}\}_{i=1,c=1,j=1}^{K,K,n_i}$ such that each row of $W^{\mathrm{mm}}$ has the following structure*

$$\boldsymbol{w}_i^{\mathrm{mm}} = \sum_{c=j=1}^{K,n_c} \alpha_{i,c,j} \boldsymbol{x}_{c,j}\,.$$

We refer the readers to Scholkopf & Smola (2018) for the proof.

# D CHARACTERIZATION OF HARD-MARGIN SVM SOLUTION FOR OTHORGONAL DATA

In this section, we characterize the hard-margin SVM solution for $\bar{K}$-class classification problem for orthogonal data. We show the solution is closely related to the $\bar{K}$-ETF simplex.

**Problem Formulation.** Consider a $\bar{K}$-class classification problem with data $\{(x_{i,j}, y_i)\}_{i=1,j=1}^{\bar{K},n}$ where:

- $\bar{K}$ is the number of classes in the dataset
- $n$ is the number of samples per class
- $y_i \in [\bar{K}]$ denotes the class label for samples $x_{i,j}$
- All data vectors $x_{i,j}$ have unit norm: $\|x_{i,j}\| = 1$
- All data vectors are mutually orthogonal: $x_{i,j}^T x_{k,l} = 0$ for $(i,j) \neq (k,l)$

Let $W \in \mathbb{R}^{\bar{K} \times d}$ be a weight matrix for a linear classifier, where the predicted class for input $x$ is $\arg\max_k (W_k x)$. For simplicity, we use the following notation to denote all the data points: $X = [x_{1,1}, \cdots, x_{\bar{K},n}]$.

Now, we are ready to present the main result.

**Theorem D.1** (Maximum Margin with Frobenius Norm). *Under the above data assumption, the largest normalized margin any linear classifier can achieve is:*

$$\gamma^* = \frac{1}{\sqrt{n(\bar{K}-1)}} \tag{18}$$

*The optimal weight matrix that achieves this margin has the form:* $W = \left( M_{\bar{K}} \otimes \mathbf{1}_n^\top \right) X^\top$.

*Proof.* Due to Proposition C.1, there exists $A^* \in \mathbb{R}^{\bar{K} \times \bar{K}n}$ such that the optimal solution $W^* = A^* X^\top$, and the corresponding logits is $W^* X = A^*$. Then, we know that $A^*$ is the solution of the following optimization problem

$$A^* \in \arg\max_{A \in \mathbb{R}^{K \times \bar{K}}} \frac{\min_{i,j \in [\bar{K}], k \in [n], j \neq i} A_{i,(i-1)n+k} - A_{j,(i-1)n+k}}{\|A\|_F} := \mathcal{L}(A). \tag{19}$$

SVM has been well studied in the literature Burges & Crisp (1999), and it has been proved that SVM has a unique solution. Then, we list several permutation invariance of the problem, and it leads to certain structures of $A^*$. We first decompose $A^*$ into block matrices, i.e., $A^* = \begin{bmatrix} A_1^* & \cdots & A_{\bar{K}}^* \end{bmatrix}$ where $A_i^* \in \mathbb{R}^{\bar{K} \times n}, i \in [\bar{K}]$.

**Invariance under column permutation within block matrices.** We first observe that one can arbitrarily permute any columns of $A^*$ within the block, and it still yields the solution to (19). Formally, for any permutation matrix $\Pi_i \in \mathbb{R}^{n \times n}$, one has $\begin{bmatrix} A_1^* \Pi_1 & \cdots & A_{\bar{K}}^* \Pi_{\bar{K}} \end{bmatrix}$ still is the solution to (19). Since (19) has a unique solution, then $A_i^* \Pi_i = A_i^*$ must hold for arbitrary permutation matrices $\Pi_i$. This leads to the conclusion that all columns of $A_i^*$ must be equal, i.e., $A_i^* = \boldsymbol{v}_i \mathbf{1}_n^\top$ for some $\boldsymbol{v}_i \in \mathbb{R}^{\bar{K}}$.

**Invariance under permutation across block matrices.** We randomly pick one column from each block, and form a matrix, i.e., $V = [\boldsymbol{v}_1, \boldsymbol{v}_2, \cdots, \boldsymbol{v}_{\bar{K}}]$. Let $\Pi' \in \mathbb{R}^{\bar{K} \times \bar{K}}$ be any permutation matrix, then one also have $\bar{V} = \Pi V \Pi^\top$ also yields a solution of (19). Specifically, let $\bar{V} = [\bar{\boldsymbol{v}}_1, \cdots, \bar{\boldsymbol{v}}_{\bar{K}}]$, one has $\begin{bmatrix} \bar{\boldsymbol{v}}_1 \mathbf{1}_n^\top & \cdots & \bar{\boldsymbol{v}}_{\bar{K}} \mathbf{1}_n^\top \end{bmatrix}$ is also a solution of (19). Based on Lemma C.5, one has $V = a\boldsymbol{I}_{\bar{K}} - b\mathbf{1}_{\bar{K}}\mathbf{1}_{\bar{K}}^\top$ where $a, b \in \mathbb{R}$.

**Scalar optimization.** Based on the above reasoning, we have concluded that $A^*$ has a simple expression which depends on two scalars, i.e., $a, b$. Now, we solve for the optimal $a, b$. we can first

derive the following closed form expression for $\mathcal{L}(A)$,

$$\mathcal{L}(A) = \frac{a}{\sqrt{n\bar{K}(a-b)^2 + n\bar{K}(\bar{K}-1)b^2}}$$

$$= \frac{1}{\sqrt{n\bar{K} + n\bar{K}^2(\frac{b}{a})^2 - 2n\bar{K}\frac{b}{a}}}$$

$$= \frac{1}{\sqrt{n}} \cdot \frac{1}{\sqrt{\bar{K}-1 + (\bar{K}\frac{b}{a}-1)^2}}$$

$$\leq \frac{1}{\sqrt{n(\bar{K}-1)}} \, ,$$

where the last inequality achieves equality when $\frac{b}{a} = \frac{1}{\bar{K}}$. $\qquad\square$

## E    PROOF OF THEOREM 3.1

In this section, we present the proof of Theorem 3.1. We first present the following functions that will be used in the proof of Theorem 3.1.

$$g_N(x) := \sqrt{\frac{\rho(K-\bar{K})}{\bar{K}(K-\bar{K})x^2 + \bar{K}K(1-x)^2}}$$

$$g_a(x) := \exp\left(\frac{g_N(x)\sqrt{\frac{\rho}{\bar{K}n}}}{\sqrt{x^2 + \frac{K}{K-\bar{K}}(1-x)^2}}\left[x^2 + \sqrt{\frac{K}{K-\bar{K}}}(1-x)^2\right]\right)$$

$$g_b(x) := \exp\left(\frac{g_N(x)\sqrt{\frac{\rho}{\bar{K}n}}}{\sqrt{x^2 + \frac{K}{K-\bar{K}}(1-x)^2}}\left[-\frac{x^2}{\bar{K}-1} + \sqrt{\frac{K}{K-\bar{K}}}(1-x)^2\right]\right)$$

$$g_c(x) := \exp\left(\frac{g_N(x)\sqrt{\frac{\rho}{\bar{K}n}}}{\sqrt{x^2 + \frac{K}{K-\bar{K}}(1-x)^2}}\left[-\frac{\bar{K}}{K-\bar{K}} \cdot \sqrt{\frac{K}{K-\bar{K}}}(1-x)^2\right]\right). \tag{20}$$

Moreover, let $x_0 \in [0,1]$ be the root of the equation

$$\frac{g_b(x)}{g_c(x)} = \frac{1}{\sqrt{\frac{K}{K-\bar{K}}} + 1} \, .$$

**Remark E.1.** *Our proof technique follows Fang et al. (2021), and the readers can see the same definition of $g_a(x), g_b(x), g_c(x)$ and $x_0$ in Fang et al. (2021). Moreover, the authors prove the existence of $x_0$ under certain conditions.*

Now, we begin the proof of Theorem 3.1.

**Theorem E.1** (Restatement of Theorem 3.1). *Let $\left(B_\lambda^*, A_\lambda^*\right)$ be any minimizer of (6). There exists a critical regularization weight $\lambda_{\mathrm{crit}} \in (0, \frac{1}{\bar{K}\sqrt{n}})$ and scalar functions $a_\lambda, b_\lambda, c_\lambda, \Theta_\lambda$ of $\lambda$ such that for any global minimizer $\left(B_\lambda^*, A_\lambda^*\right)$ of (6) with $\lambda > 0$, we have*

$$B_\lambda^* A_\lambda^* = \left(\left(\begin{matrix}(a_\lambda + b_\lambda)\,\mathbf{I}_{\bar{K}} - b_\lambda\,\mathbf{1}_{\bar{K}}\mathbf{1}_{\bar{K}}^\top \\ -c_\lambda\,\mathbf{1}_{K-\bar{K}}\mathbf{1}_{\bar{K}}^\top\end{matrix}\right) \otimes \mathbf{1}_n^\top\right) X_{\mathrm{ft}}^\top \, , \quad \|B_\lambda^* A_\lambda^*\|_F = \Theta_\lambda \, . \tag{21}$$

*When $\bar{K} \geq 2$, the scalar functions $a_\lambda, b_\lambda, c_\lambda, \Theta_\lambda$ are characterized as follows:*

*(i) High-penalty regime $\left(\lambda \geq \frac{1}{\bar{K}\sqrt{n}}\right)$: $a_\lambda = b_\lambda = c_\lambda = \Theta_\lambda = 0$, thus $B_\lambda^* A_\lambda^* = \mathbf{0}_{K \times d}$.*

*(ii) Intermediate regime $\left(\lambda_{\mathrm{crit}} < \lambda < \frac{1}{\bar{K}\sqrt{n}}\right)$: $a_\lambda = \frac{\Theta_\lambda}{\sqrt{\bar{K}n}}, b_\lambda = \frac{\Theta_\lambda}{\bar{K}\sqrt{\bar{K}n}}, c_\lambda = 0$, and $\Theta_\lambda$ is the unique root of a nonlinear equation (see (28) in Appendix E), thus the minimizer is*

$$B_\lambda^* A_\lambda^* = \frac{\Theta_\lambda}{\sqrt{\bar{K}n}}\left(\left(\begin{matrix}M_{\bar{K}} \\ \mathbf{0}_{(K-\bar{K})\times\bar{K}}\end{matrix}\right) \otimes \mathbf{1}_n^\top\right) X_{\mathrm{ft}}^\top \, . \tag{22}$$

*(iii) Low-penalty regime $\left(\lambda \leq \lambda_{\mathrm{crit}}\right)$: in general $a_\lambda, b_\lambda, c_\lambda, \Theta_\lambda$ are positive. Nevertheless,*

$$\lim_{\lambda \to 0^+} \Theta_\lambda = \infty, \qquad \lim_{\lambda \to 0^+} \frac{B_\lambda^* A_\lambda^*}{\|B_\lambda^* A_\lambda^*\|_F} = \sqrt{\frac{1}{nK}} \left(M_K^{(\bar{K})} \otimes \mathbf{1}_n^\top\right) X_{\mathrm{ft}}^\top, \tag{23}$$

*where $K$ is the number of pre-training classes and $M_K^{(\bar{K})}$ is the first $\bar{K}$ columns of the K-ETF matrix $M_K$.*

*Proof.* We first show that one can simplify the objective in (6) based on the following Lemma.

**Lemma E.1.** *Under Assumptions 2.3 and 2.4, the original fine-tuning problem in (6) is equivalent to the following symmetric formulation:*

$$\min_{B,\tilde{A}} \mathcal{L}(B, \tilde{A}) := \frac{1}{\bar{K}n} \sum_{c=1}^{\bar{K}} \sum_{j=1}^{n} \mathcal{L}_{\mathrm{CE}}\big(y_c, B\tilde{A}e_{n(c-1)+j}\big) + \frac{\lambda}{2}\big(\|B\|_F^2 + \|\tilde{A}\|_F^2\big), \tag{24}$$

*where $X_{\mathrm{ft}} = [\bar{x}_{1,1}, \ldots, \bar{x}_{1,n}, \cdots, \bar{x}_{\bar{K},1}, \cdots \bar{x}_{\bar{K},n}], B \in \mathbb{R}^{K \times r}, \tilde{A} = AX_{\mathrm{ft}} \in \mathbb{R}^{r \times \bar{K}n}$.*

*Let $X_{\mathrm{ft},\perp}$ be an orthonormal complement of $X_{\mathrm{ft}}$. Then:*

1. *If $(B^*, A^*)$ is any global minimizer of (6), then*

$$(B^*, \tilde{A}^*) \quad with \quad \tilde{A}^* = A^* X_{\mathrm{ft}}$$

   *is a global minimizer of (24).*

2. *Conversely, if $(B^*, \tilde{A}^*)$ is any global minimizer of (24), then $(B^*, \tilde{A}^* X_{\mathrm{ft}}^\top)$ yields a global minimizer of the original problem in (6).*

We refer the readers to Appendix F.1 for the proof of Lemma E.1. Lemma E.1 indicates that to characterize the global minimizer of the objective in (6), one can instead characterize the global minimizer of the objective in (24).

We start with the following equations that characterize the global minimizer of (24)

$$\frac{\partial \mathcal{L}(B, \tilde{A})}{\partial B} = \lambda B + \frac{1}{\bar{K}n} \sum_{c=1}^{\bar{K}} \sum_{j=1}^{n} \frac{\partial}{\partial B\tilde{A}}\left(\mathcal{L}_{\mathrm{CE}}\left(y_c, B\tilde{A}e_{(c-1)n+j}\right)\right) e_{(c-1)n+j}^\top \tilde{A}^\top = 0$$

$$\frac{\partial \mathcal{L}(B, \tilde{A})}{\partial \tilde{A}} = \lambda \tilde{A} + B^\top \frac{1}{\bar{K}n} \sum_{c=1}^{\bar{K}} \sum_{j=1}^{n} \frac{\partial}{\partial B\tilde{A}}\left(\mathcal{L}_{\mathrm{CE}}\left(y_c, B\tilde{A}e_{(c-1)n+j}\right)\right) e_{(c-1)n+j}^\top = 0.$$

Based on the above equation, we can conclude that any minimizer $(B, \tilde{A})$ of (24) must satisfy

$$\lambda B^\top B = \lambda \tilde{A}\tilde{A}^\top \Rightarrow \|B\|_F^2 = \|\tilde{A}\|_F^2. \tag{25}$$

Our second step is to converge the problem into a constraint optimization problem. The following lemma characterizes this.

**Lemma E.2.** *Let $B \in \mathbb{R}^{K \times r}, \tilde{A} \in \mathbb{R}^{r \times \bar{K}}$. For each $\lambda > 0$, there exists a unique non-negative value $\rho_\lambda$ such that the solution set for the following optimization problems are equivalent*

$$\min_{B,\tilde{A}} \frac{1}{\bar{K}n} \sum_{c=1}^{\bar{K}} \sum_{j=1}^{n} \mathcal{L}_{\mathrm{CE}}\big(y_c, B\tilde{A}e_{(c-1)n+j}\big) + \frac{\lambda}{2}\big(\|B\|_F^2 + \|\tilde{A}\|_F^2\big), \qquad \text{(Problem one)}$$

$$\min_{B,\tilde{A}} \frac{1}{\bar{K}n} \sum_{c=1}^{\bar{K}} \sum_{j=1}^{n} \mathcal{L}_{\mathrm{CE}}\big(y_c, B\tilde{A}e_{(c-1)n+j}\big) \quad s.t. \|B\|_F^2 \leq \rho_\lambda, \|\tilde{A}\|_F^2 \leq \rho_\lambda. \qquad \text{(Problem two)}$$

*Moreover, the map $\lambda \mapsto \rho_\lambda$ enjoys the following properties:*

*(i) **Monotonicity and continuity.** $\rho_\lambda$ is continuous and non-increasing on $(0, \infty)$.*

*(ii)* ***Asymptotic behaviour.***

$$\lim_{\lambda \to 0^+} \rho_\lambda = \infty, \qquad \lim_{\lambda \to \infty} \rho_\lambda = 0.$$

*(iii)* ***Flat regions.*** *For any* $0 < \lambda_1 < \lambda_2$, $\rho(\lambda_1) = \rho(\lambda_2) \iff \rho(\lambda_1) = \rho(\lambda_2) = 0$.

We refer the readers to Appendix F.2 for the proof. For simplicity, we will use $\rho$ to denote $\rho_\lambda$ for convenience.

**Characterization to solution of** (Problem two). (Problem two) can be viewed as a special case of the neural collapse phenomenon in the extreme setting of an imbalanced dataset: for the first $\bar{K}$ classes we observe $n$ data points per class, while all remaining classes have zero samples. In this context, we directly invoke the following result from Fang et al. Fang et al. (2021) (see p. 26 in . Fang et al. (2021)).

**Lemma E.3** (Neural collapse under extreme imbalance; (Fang et al., 2021, Page 26)). *For any constants* $c_1, c_2, c_3, \rho > 0$, *define*

$$c_1' = \frac{c_1}{c_1 + (\bar{K} - 1)c_2 + (K - \bar{K})c_3}$$

$$c_2' = \frac{c_2}{c_1 + (\bar{K} - 1)c_2 + (K - \bar{K})c_3}$$

$$c_3' = \frac{c_3}{c_1 + (\bar{K} - 1)c_2 + (K - \bar{K})c_3}$$

$$c_4 = -c_1' \log c_1' - c_2'(\bar{K} - 1) \log c_2' - (K - \bar{K})c_3' \log(c_3')$$

$$c_5 = \frac{\bar{K}c_2}{\bar{K}c_2 + (K - \bar{K})c_3}$$

$$c_6 = \frac{(K - \bar{K})c_3}{\bar{K}c_2 + (K - \bar{K})c_3}$$

$$c_7 = \frac{\bar{K}c_2 + (K - \bar{K})c_3}{c_1 + (\bar{K} - 1)c_2 + (K - \bar{K})c_3}$$

*For any feasible solution* $(B, \tilde{A})$ *of* (Problem two)*, the objective value of* (Problem two) *can be simplified to*

$$\frac{1}{\bar{K}n} \sum_{c=1}^{\bar{K}} \sum_{j=1}^{n} \mathcal{L}_{CE}\left(y_c, B\tilde{A}e_{(c-1)n+j}\right) \overset{a}{\geq} -\frac{c_7}{\bar{K}}\sqrt{\frac{\rho}{n}}\sqrt{\sum_{i=1}^{\bar{K}} \|c_5 w_1 + c_6 w_2 - b_i\|^2} + c_4$$

*where* $w_1 = \frac{1}{\bar{K}} \sum_{i=1}^{\bar{K}} b_i$, $w_2 = \frac{1}{K - \bar{K}} \sum_{i=\bar{K}+1}^{K} b_i$, *and* $\overset{a}{\geq}$ *becomes equality under certain choices of* $c_1, c_2, c_3$.

Readers are referred to Fang et al. (2021) for the full proof. Note the following distinction between Lemma E.3 and its analogue in Fang et al. (2021): the inequality in Fang et al. (2021) becomes tight only when all minority-class features are zero and certain choices of $c_1, c_2, c_3$, whereas in our lemma, we do not require that all minority-class features are zero. This difference is resolved by our change of variables $\tilde{A} = AX_{ft}$, which effectively confines $\tilde{A}$ to the subspace of observed (majority-class) features and thus guarantees equality in Lemma E.3.

Now, we carefull pick the $B, \tilde{A}$ and $c_1, c_2, c_3$ to achieve the global minimum of the objective in (Problem two). The following lemma characterizes this exactly.

**Lemma E.4** (Lemma 5 in Fang et al. (2021)). *Under the same assumptions in Theorem 3.1, there exists a*

$$\rho_{crit} = \sqrt{n}\bar{K}(\bar{K}-1) \log\left(\sqrt{\frac{K}{K - \bar{K}}} + 1\right)$$

*such that the optimal value to* (Problem two) *is as follows*

- *when $\bar{K} \geq 2$ and $\rho_\lambda < \rho_{\mathrm{crit}}$, we choose $c_1 = \exp\left(\frac{\rho_\lambda}{\bar{K}\sqrt{n}}\right), c_2 = \exp\left(\frac{-\rho_\lambda}{\sqrt{n}\bar{K}(\bar{K}-1)}\right), c_3 = 1,$
  and* (Problem two) *attain its minimum*

$$\lambda\rho_\lambda - \frac{\rho_\lambda}{\sqrt{n}\bar{K}} + \log\left(K - \bar{K} + (\bar{K}-1)\exp\left(-\frac{\rho_\lambda}{\sqrt{n}\bar{K}(\bar{K}-1)}\right) + \exp\left(\frac{\rho_\lambda}{\sqrt{n}\bar{K}}\right)\right),$$

  *where $B^*, \tilde{A}^*$ takes the following form*

$$\left[b_1^*, b_2^*, \ldots, b_{\bar{K}}^*\right] = \sqrt{n}\left[\bar{a}_1^*, \bar{a}_2^*, \ldots, \bar{a}_{\bar{K}}^*\right] = \sqrt{\frac{\rho}{K}}\, M_n,$$

$$a_i^* = \bar{a}_{\lceil i/n\rceil}^*, \quad i = 1, \ldots, \bar{K}n,$$
$$b_j^* = 0, \qquad j > \bar{K}.$$

  *Moreover, $B^*\tilde{A}^*$ takes the following form*

$$B^* A^* = \frac{\rho_\lambda}{\sqrt{\bar{K}n(\bar{K}-1)}}\begin{pmatrix} M_{\bar{K}} \\ \mathbf{0}_{(K-\bar{K})\times\bar{K}} \end{pmatrix} \otimes \mathbf{1}_n^\top. \tag{26}$$

- *when $\bar{K} \geq 2$ and $\rho \geq \rho_{\mathrm{crit}}$, one choose*
$$c_1 = g_a(x_0), \quad c_2 = g_b(x_0), \quad c_3 = g_c(x_0),$$
  *where $g_a(x), g_b(x), g_c(x)$ and $x_0$ are defined in* (20). *Then,* (Problem two) *attain its minimum*

$$\lambda\rho_\lambda + \log\left(\frac{g_a(x_0) + (\bar{K}-1)g_b(x_0) + (K-\bar{K})g_c(x_0)}{g_a(x_0)}\right),$$

  *where*

$$b_i^* = \begin{cases} g_N(x_0) P_A\left[\frac{\bar{K}x_0}{\sqrt{\bar{K}(\bar{K}-1)}}\bar{y}_i + \left(\frac{1-x_0}{\sqrt{\bar{K}}} - \frac{x_0}{\sqrt{\bar{K}(\bar{K}-1)}}\right)\mathbf{1}_{\bar{K}}\right], & i \leq \bar{K}, \\ -\frac{g_N(x_0)\sqrt{\bar{K}}(1-x_0)}{K-\bar{K}}P\mathbf{1}_{\bar{K}}, & i > \bar{K}. \end{cases}$$

$$\bar{a}_i^* = \begin{cases} g_N(x_0)P\left[x_0\sqrt{\frac{\bar{K}}{\bar{K}-1}}\bar{y}_c + \left(\frac{(1-x_0)g_N(x_0)\sqrt{\frac{K}{K-\bar{K}}}}{\sqrt{\bar{K}}} - \frac{g_N(x_0)x_0}{\sqrt{\bar{K}(\bar{K}-1)}}\right)\mathbf{1}_{\bar{K}}\right], & c-1 \leq \frac{i}{n} < c, \\ 0, & i > n\bar{K}. \end{cases}$$

  *where $\bar{y}_c \in \mathbb{R}^{\bar{K}}$ and is a one-hot vector with $i$-th entry equals one, and $P \in \mathbb{R}^{r\times\bar{K}}$ is a partial orthogonal matrix such that $P^\top P = \mathbf{I}_{\bar{K}}$. Moreover, $B^*\tilde{A}^*$ takes the following form*

$$B^*\tilde{A}^* = \begin{pmatrix} (\log g_a(x_0) + \log g_b(x_0))\mathbf{I}_{\bar{K}} - \log g_b(x_0)\mathbf{1}_{\bar{K}}\mathbf{1}_{\bar{K}}^\top \\ \log g_c(x_0)\mathbf{1}_{K-\bar{K}}\mathbf{1}_{\bar{K}}^\top \end{pmatrix} \otimes \mathbf{1}_n^\top.$$

Lemma E.4 is Lemma 5 in Fang et al. (2021). We refer the readers to Fang et al. (2021) for the proof. Lemma E.4 exactly characterize the optimal solution and value of the objective in (Problem two).

**Computation of the product of LoRA adapters.** Based on Lemma E.4, we can characterize the solution to the LoRA objective in (6), then it suffices to compute the product of the minimizer to prove Theorem 3.1.

In the **Intermediate regime**, one can first compute

$$B^*\left[\bar{a}_1^*, \bar{a}_2^*, \ldots, \bar{a}_{\bar{K}}^*\right] = \frac{\rho_\lambda}{\bar{K}\sqrt{n}}M_n^\top M_n$$

$$= \frac{\rho_\lambda}{\bar{K}\sqrt{n}}\frac{\bar{K}}{\bar{K}-1}\left(\mathbf{I}_{\bar{K}} - \frac{1}{\bar{K}}\mathbf{1}_{\bar{K}}\mathbf{1}_{\bar{K}}^\top\right)$$

$$= \frac{\rho_\lambda}{\sqrt{\bar{K}n(\bar{K}-1)}}M_{\bar{K}},$$

and it leads to the product of final solution

$$B^* \tilde{A}^* = \frac{\rho_\lambda}{\sqrt{\bar{K} n (\bar{K} - 1)}} \begin{pmatrix} M_{\bar{K}} \\ \mathbf{0}_{(K-\bar{K}) \times \bar{K}} \end{pmatrix} \otimes \mathbf{1}_n^\top . \tag{27}$$

In the **Low penalty regime**, in Fang et al. (2021), the authors further show that

$$(b_i^*)^\top \bar{a}_j^* = \begin{cases} \log g_a(x_0) & i = j \leq \bar{K} \\ \log g_b(x_0) & i \neq j, i \leq \bar{K} \\ \log g_c(x_0) & \bar{K} < i \leq K \end{cases}$$

Thus, one can show that

$$B^* \tilde{A}^* = \begin{pmatrix} (\log g_a(x_0) + \log g_b(x_0)) \mathbf{I}_{\bar{K}} - \log g_b(x_0) \mathbf{1}_{\bar{K}} \mathbf{1}_{\bar{K}}^\top \\ \log g_c(x_0) \mathbf{1}_{K-\bar{K}} \mathbf{1}_{\bar{K}}^\top \end{pmatrix} \otimes \mathbf{1}_n^\top .$$

**Characterization of $\lambda_{\mathrm{crit}}$.** In Lemma E.4, when $m \geq 2$, we see that the solution takes different form based on whether $\rho_\lambda < \rho_{\mathrm{crit}}$ or $\rho_\lambda \geq \rho_{\mathrm{crit}}$. Moreover, based on Lemma E.2, there must exist a unique $\lambda_{\mathrm{crit}}$ such that $\rho_{\mathrm{crit}} = \rho(\lambda_{\mathrm{crit}})$. Now, we characterize the range of $\lambda_{\mathrm{crit}}$ here.

Due to Lemma E.4, we know when $\lambda > \lambda_{\mathrm{crit}}$, the solution takes the form

$$B^* \tilde{A}^* = \begin{pmatrix} \frac{\rho}{\sqrt{nK}} \mathbf{I}_{\bar{K}} \\ \mathbf{0}_{(K-\bar{K}) \times \bar{K}} \end{pmatrix} \otimes \mathbf{1}_n^\top ,$$

and the corresponding minimum objective is

$$\psi(\rho) := \lambda \rho - \frac{\rho}{\sqrt{n\bar{K}}} + \log \left( K - \bar{K} + (\bar{K} - 1) \exp \left( - \frac{\rho}{\sqrt{n\bar{K}(\bar{K} - 1)}} \right) + \exp \left( \frac{\rho}{\sqrt{n\bar{K}}} \right) \right),$$

Now, we take the derivative of $\psi(\rho)$

$$\frac{d\psi(\rho)}{d\rho} = 0$$

$$\iff \left( \lambda - \frac{1}{\sqrt{n\bar{K}}} \right) + \frac{1}{\sqrt{n\bar{K}}} \cdot \frac{\exp\left(\frac{\rho}{\sqrt{n\bar{K}}}\right) - \exp\left(-\frac{\rho}{\sqrt{n\bar{K}(\bar{K}-1)}}\right)}{K - \bar{K} + (\bar{K}-1)\exp\left(-\frac{\rho}{\sqrt{n\bar{K}(\bar{K}-1)}}\right) + \exp\left(\frac{\rho}{\sqrt{n\bar{K}}}\right)} = 0$$

$$\iff \left( \lambda - \frac{1}{\sqrt{n\bar{K}}} \right) + \frac{1}{\sqrt{n\bar{K}}} \cdot \frac{\exp\left(\frac{\rho}{\sqrt{n(\bar{K}-1)}}\right) - 1}{(K - \bar{K})\exp\left(\frac{\rho}{\sqrt{n\bar{K}(\bar{K}-1)}}\right) + (\bar{K}-1) + \exp\left(\frac{\rho}{\sqrt{n(\bar{K}-1)}}\right)} = 0 \tag{28}$$

Based on (28), one know if $\lambda_{\mathrm{crit}} > \frac{1}{\sqrt{n\bar{K}}}$, then $\frac{d\psi(\rho)}{d\rho} > 0$, and the minimum is achieved when $\rho_\lambda = 0$. On the other hand, we know that when $\lambda < \lambda_{\mathrm{crit}}$, one has $\rho_\lambda \geq \rho_{\mathrm{crit}}$. Thus, $\rho_{\mathrm{crit}}$ is a dis-continuous point of $\rho_\lambda$. However, based on Lemma E.2, we know $\rho_\lambda$ is a continuous function. Thus, one must have

$$\lambda_{\mathrm{crit}} < \frac{1}{\sqrt{n\bar{K}}} .$$

**Characterization of $\rho_\lambda$.** In this part, we characterize the $\rho_\lambda$ when $\lambda > \lambda_{\mathrm{crit}}$. Based on Lemma E.4, we know $\psi(\rho)$ is the optimal value of (Problem one) and (Problem two). Thus, $\rho_\lambda$ must attain the minimum of $\psi(\rho)$. Based on (28), one can first see that if $\lambda \geq \frac{1}{\sqrt{n\bar{K}}}$, $\frac{d\psi(\rho)}{d\rho} > 0$, and the minimum is attained when $\rho_\lambda = 0$. When $\lambda_{\mathrm{crit}} < \lambda < \frac{1}{\sqrt{n\bar{K}}}$, one can solve let $\frac{d\psi(\rho)}{d\rho} = 0$ in (28) to seek for $\rho_\lambda$.

**Asymptotic behaviour of $\rho_\lambda$.** On one hand, as $\lambda \to 0+$, one obviously has $\lim_{\lambda \to 0+} \|B^* \tilde{A}^*\|_F = \infty$ due to the fact that $B^* \tilde{A}^*$ is the solution of (Problem three). Then, we characterize its asymptotic direction, which is equivalent to study the limit of $\frac{\log g_a(x_0)}{\log g_b(x_0)}$ and $\frac{\log g_a(x_0)}{\log g_c(x_0)}$ as $\lambda \to 0+$. We will use $x_{0,\lambda}$ denote the choice of $x_0$ for fix regularization effect $\lambda$.

First, the equation to derive for $x_0$ is as follows

$$\frac{g_b(x)}{g_c(x)} = \frac{1}{1 + \sqrt{\frac{K}{K - \bar{K}}}}$$

$$\Longleftrightarrow \frac{\rho_\lambda}{\bar{K}\sqrt{n}} \cdot \frac{-\frac{x^2}{\bar{K}-1} + (\frac{K}{K-\bar{K}})^{3/2}(1-x)^2}{x^2 + \frac{K}{K-\bar{K}}(1-x)^2} = -\log\big(1 + \sqrt{\frac{K}{K - \bar{K}}}\big)$$

Since $\lim_{\lambda \to 0+} \rho_\lambda = \infty$, one must have $x_{0,\lambda}$ approaches the solution of the following quadratic equation

$$\frac{x_{0,\lambda}^2}{\bar{K} - 1} = \big(\frac{K}{K - \bar{K}}\big)^{3/2}(1 - x_{0,\lambda})^2 .$$

Moreover, one can see that the expression for $\frac{\log g_a(x_{0,\lambda})}{\log g_b(x_{0,\lambda})}$ and $\frac{\log g_a(x_{0,\lambda})}{\log g_c(x_{0,\lambda})}$ takes the following form

$$\frac{\log g_a(x_{0,\lambda})}{\log g_b(x_{0,\lambda})} = \frac{x_{0,\lambda}^2 + \sqrt{\frac{K}{K-\bar{K}}}(1 - x_{0,\lambda})^2}{-\frac{x_{0,\lambda}^2}{\bar{K}-1} + \sqrt{\frac{K}{K-\bar{K}}}(1 - x_{0,\lambda})^2}$$

$$\frac{\log g_a(x_{0,\lambda})}{\log g_c(x_{0,\lambda})} = \frac{x_{0,\lambda}^2 + \sqrt{\frac{K}{K-\bar{K}}}(1 - x_{0,\lambda})^2}{-\frac{\bar{K}}{K-\bar{K}} \cdot \sqrt{\frac{K}{K-\bar{K}}}(1 - x_{0,\lambda})^2} .$$

Thus, combine the above two equations, we can compute that

$$\lim_{\lambda \to 0+} \frac{\log g_a(x_{0,\lambda})}{\log g_b(x_{0,\lambda})} = \lim_{\lambda \to 0+} \frac{\log g_a(x_{0,\lambda})}{\log g_c(x_{0,\lambda})} = -(K - 1) .$$

**Connection between $\rho_\lambda$ and $\Theta_\lambda$.** In previous analysis, we have shown that

$$\rho_\lambda = \|B_\lambda^* A_\lambda^*\|_* , \qquad \Theta_\lambda = \|B_\lambda^* A_\lambda^*\|_F \tag{29}$$

Thus, what is left is to show that under different regularization paramter, what is the relation between $\rho_\lambda$ and $\Theta_\lambda$.

First, it is obvious in the high-penalty regime, $\rho_\lambda = \Theta_\lambda = 0$.

Second, in the *intermediate regime*, the product of the optimal LoRA adapters are

$$B^* A^* = \frac{\rho_\lambda}{\sqrt{\bar{K}n(\bar{K}-1)}} \begin{pmatrix} M_{\bar{K}} \\ \mathbf{0}_{(K-\bar{K})\times\bar{K}} \end{pmatrix} \otimes \mathbf{1}_n^\top .$$

After some simple calculations, one can show that

$$\rho_\lambda = \|B_\lambda^* A_\lambda^*\|_* , \qquad \Theta_\lambda := \|B_\lambda^* A_\lambda^*\|_F = \frac{\rho_\lambda}{\sqrt{\bar{K}-1}}$$

Thus, one can alternatively represent the product of the optimal LoRA adapters in the *intermediate regime* as

$$B^* A^* = \frac{\Theta_\lambda}{\sqrt{n(\bar{K}-1)}} \begin{pmatrix} M_{\bar{K}} \\ \mathbf{0}_{(K-\bar{K})\times\bar{K}} \end{pmatrix} \otimes \mathbf{1}_n^\top .$$

Third, in the *low-penalty regime*, one can first see that $g_a, g_b, g_c$ go to infinity as the $\lambda \to 0_+$, thus, $\lim_{\lambda \to 0_+} \|B^* \tilde{A}^*\|_F = \infty$. Then, the directions of the limiting behaviour is characterized right above.

**The solution of** (6). Finally, one can use Lemma E.1 to characterize the optimal solution of (6) based on the above results.

$\square$

## F  PROOF OF SEVERAL LEMMAS

### F.1  PROOF OF LEMMA E.1

*Proof.* We first observe that based on Proposition C.1, Assumption 2.3 and Assumption 2.4, one has

$$W_{\mathrm{pre}}X_{\mathrm{ft}} = 0\,. \tag{30}$$

Moreover, under Assumption 2.3, we have $X_{\mathrm{ft}}^\top X_{\mathrm{ft}} = \boldsymbol{I}_{\bar{K}n}$. We apply change of variable as follows: $\tilde{A} = AX_{\mathrm{ft}} \in \mathbb{R}^{r \times \bar{K}n}$.

Under this transformation, the objective in (6) can be rewritten as follows

$$\frac{1}{\bar{K}n}\sum_{c=1}^{\bar{K}}\sum_{j=1}^{n}\mathcal{L}_{\mathrm{CE}}\left(y_c, (W_{\mathrm{pre}} + BA)x_{c,j}\right) + \frac{\lambda}{2}\left(\|B\|_F^2 + \|A\|_F^2\right)$$

$$=\frac{1}{\bar{K}n}\sum_{c=1}^{\bar{K}}\sum_{j=1}^{n}\mathcal{L}_{\mathrm{CE}}\left(y_c, BAX_{\mathrm{ft}}e_{(c-1)n+j}\right) + \frac{\lambda}{2}\left(\|B\|_F^2 + \|A\|_F^2\right)$$

$$=\frac{1}{\bar{K}n}\sum_{c=1}^{\bar{K}}\sum_{j=1}^{n}\mathcal{L}_{\mathrm{CE}}\left(y_c, B\tilde{A}e_{(c-1)n+j}\right) + \frac{\lambda}{2}\left(\|B\|_F^2 + \|AX_{\mathrm{ft}}\|_F^2 + \|AX_{\mathrm{ft},\perp}\|_F^2\right)$$

$$=\frac{1}{\bar{K}n}\sum_{c=1}^{\bar{K}}\sum_{j=1}^{n}\mathcal{L}_{\mathrm{CE}}\left(y_c, B\tilde{A}e_{(c-1)n+j}\right) + \frac{\lambda}{2}\left(\|B\|_F^2 + \|\tilde{A}\|_F^2 + \|AX_{\mathrm{ft},\perp}\|_F^2\right)\,.$$

Notice that the component $AX_{\mathrm{ft},\perp}$ contributes only to the *regularization* term and does not affect the cross-entropy loss. Hence at any global minimizer $(B^*, A^*)$ of (6) we must have

$$AX_{\mathrm{ft},\perp} = 0.$$

Moreover, if $(B^*, A^*)$ is a global minimizer of the original problem, then

$$(B^*,\ A^*X_{\mathrm{ft}})$$

is a global minimizer of the objective in (24). Conversely, let $(B^*, \tilde{A}^*)$ be any global minimizer of (24), then one can solve the following equation to obtain $(B^*, A^*)$ which is a global minimizer of the original objective in (6).

$$A^*X_{\mathrm{ft}} = \tilde{A}^*, \quad A^*X_{\mathrm{ft},\perp} = 0\,. \tag{31}$$

$\square$

### F.2  PROOF OF LEMMA E.2

*Proof.* To show that there exists a unique value $\rho_\lambda$ such that the (Problem one) and (Problem two) enjoys the same set of solutions, we first introduce two additional optimization problems.

$$\min_{Z}\ \frac{1}{\bar{K}n}\sum_{c=1}^{\bar{K}}\sum_{j=1}^{n}\mathcal{L}_{\mathrm{CE}}\left(y_c,\ Ze_{(c-1)n+j}\right) + \lambda\|Z\|_*\,, \tag{Problem three}$$

$$\min_{Z}\ \frac{1}{\bar{K}n}\sum_{c=1}^{\bar{K}}\sum_{j=1}^{n}\mathcal{L}_{\mathrm{CE}}\left(y_c,\ Ze_{(c-1)n+j}\right) \quad \mathrm{s.t.}\|Z\|_* \le \rho_\lambda\,. \tag{Problem four}$$

where $Z \in \mathbb{R}^{K \times \bar{K}n}$. Let $S_i(\lambda), i = 1, 2, 3, 4$ be the solution sets of the above four optimization problems for the same fixed $\lambda$. We will show that

- $S_1(\lambda) = S_2(\lambda), S_3(\lambda) = S_4(\lambda)$.

- $S_3(\lambda), S_4(\lambda)$ contains only one element for any fixed $\lambda > 0$.

- $\forall B^*, \tilde{A}^* \in S_1(\lambda)$, one has $B^* \tilde{A}^* \in S_3(\lambda)$.

- Let $S_3(\lambda) = S_4(\lambda) = \{Z_\lambda^*\}$, then there exists $B^* \tilde{A}^* = Z_\lambda^*$ such that $(B^*, \tilde{A}^*) \in S_1(\lambda)$.

Based on Lemma C.2, one can see that (Problem three) is a convex version of (Problem one) in the sense that $\forall (B^*, \tilde{A}^*) \in S_1(\lambda)$, $B^* \tilde{A}^* \in S_3(\lambda)$. Moreover, $\forall Z^* \in S_3(\lambda)$, let the SVD of $Z^*$ be $Z^* = U\Sigma V^\top$. Then, $B^* = U\Sigma^{1/2}$, $\tilde{A}^* = \Sigma^{1/2}V^\top$ is also a solution for (Problem one). Generally speaking, $\forall (B^*, \tilde{A}^*) \in S_1(\lambda)$ must satisfy

$$2\|B^* \tilde{A}^*\|_* = \|B^*\|_F^2 + \|\tilde{A}^*\|_F^2.$$

Now we show (Problem three) has a unique solution, and it leads to $S_3(\lambda) = S_4(\lambda)$ with certain choices of $\rho_\lambda$. Our analysis is based on the following lemma.

**Lemma F.1** (Cross entropy loss with nuclear norm has a unique minimizer). *For any $\lambda > 0$, define*

$$\phi(Z; \lambda) = \frac{1}{\bar{K}n} \sum_{c=1}^{\bar{K}} \sum_{j=1}^{n} \mathcal{L}_{\text{CE}}\big(y_c,\, Ze_{(c-1)n+j}\big) + \lambda\|Z\|_*,$$

*where $y_c$ is a one-hot vector with $c$-th index equals one. Then $\phi(Z; \lambda)$ has a unique solution.*

We refer the readers to later sections in Appendix F.3 for the proof.

Based on Lemma F.1, (Problem three) has a unique solution, denoted by $Z_\lambda^*$. Then, one can choose $\rho_\lambda = \|Z_\lambda^*\|_*$, and based on strong duality, (Problem three) and (Problem four) admits the same set of solutions, i.e., $S_3(\lambda) = S_4(\lambda)$.

Next, we show that $\forall (B^*, \tilde{A}^*) \in S_2(\lambda)$, it must satisfy

$$B^* \tilde{A}^* \in S_4(\lambda), \qquad 2\|B^* \tilde{A}^*\|_* = \|B^*\|_F^2 + \|\tilde{A}^*\|_F^2.$$

This is because first, the minimum objective value of (Problem two) and (Problem four) must be equal. Given $Z_\lambda^* \in S_4(\lambda)$, one can do balanced factorization, and obtain the corresponding $B^*, \tilde{A}^*$. This implies the minimum objective value of (Problem two) must be larger or equal than the one of (Problem four). On the other hand, $\forall (B^*, \tilde{A}^*) \in S_2(\lambda)$, one has $\|B^* \tilde{A}^*\|_* \leq \frac{\|B^*\|_F^2 + \|\tilde{A}^*\|_F^2}{2} \leq \rho_\lambda$ which implies $B^* \tilde{A}^*$ is a feasible solution of (Problem four). Thus, this further implies the minimum objective value of (Problem four) must be larger or equal than the one of (Problem two). Combine these together, we conclude that the minimum objective value of (Problem two) and (Problem four) must be equal, and $\forall (B^*, \tilde{A}^*) \in S_2(\lambda)$, $B^* \tilde{A}^* \in S_4(\lambda)$ must hold.

Finally, we show $S_1(\lambda) = S_2(\lambda)$.

This is because, on one hand, $\forall B_1^*, \tilde{A}_1^* \in S_1(\lambda)$, from previous reasoning, they must satisfy

$$B_1^* \tilde{A}_1^* \in S_3(\lambda) = S_4(\lambda), \qquad \|B_1^*\|_F^2 = \|\tilde{A}_1^*\|_F^2 = \rho_\lambda,$$

and this indicates $B_1^*, \tilde{A}_1^* \in S_2(\lambda)$.

On the other hand, $\forall B_2^*, \tilde{A}_2^* \in S_2(\lambda)$, they must satisfy

$$B_2^* \tilde{A}_2^* \in S_3(\lambda) = S_4(\lambda), \qquad \|B_2^*\|_F^2 = \|\tilde{A}_2^*\|_F^2 = \rho_\lambda,$$

and this indicates $B_2^*, \tilde{A}_2^* \in S_2(\lambda)$. Thus, we finish the proof for $S_1(\lambda) = S_2(\lambda)$.

The last step is to show $\rho_\lambda$ is a continuous and non-increasing function of $\lambda$ that satisfies

$$\lim_{\rho \to 0+} \rho_\lambda = \infty, \qquad \lim_{\rho \to \infty} \rho_\lambda = 0.$$

**Continuity of $\rho_\lambda$.** In this part, we will use Lemma C.4 to prove the solution to (Problem three) is continuous, which implies that $\rho_\lambda$ is a continuous function w.r.t. $\lambda$. Notice Lemma C.4 is presented for maximization problem, one can replace the objective $f$ to $-f$ to extend it to minimization problem. Moreover, it is obvious that the objective functon is continuous, convex, and the set of minimizers is single-valued for any fixed $\lambda > 0$. Thus, in order to apply Lemma C.4 to show the continuity

of $\rho_\lambda$, we only need to check $C(\lambda)$, which is the range of $Z$, is a compact-valued correspondence. At first glance, this seems wrong since $Z$ can take any values in $\mathbb{R}^{K \times \tilde{K}n}$ in (Problem three). However, we will show that one can constrain the domain of $Z$ which leads to the same minimizer. Notice that when $Z = \mathbf{0}$, the objective in (Problem three) takes the value $\log K$. Thus, we can choose $C(\lambda) = \left\{ Z \,\middle|\, \|Z\|_* \leq \frac{\log K}{\lambda} \right\}$, which is a compact set. Therefore, for any fix $\lambda$, one can apply Lemma C.4 to show $Z_\lambda^*$ is continuous w.r.t. $\lambda$, which implies $\rho_\lambda := \|Z_\lambda^*\|_*$ is also continuous w.r.t. $\lambda$.

$\rho_\lambda$ **is a non-increasing function.** For any $0 < \lambda_1 < \lambda_2$, let $Z_{\lambda_1}, Z_{\lambda_2}$ be the solution for (Problem three) with the corresponding regularization penalty. Then, our first observation is

$$\phi(Z_{\lambda_2}; \lambda_2) \leq \phi(Z_{\lambda_1}; \lambda_2)$$
$$\iff \phi(Z_{\lambda_2}; \lambda_1) + (\lambda_2 - \lambda_1)\|Z_{\lambda_2}\|_* \leq \phi(Z_{\lambda_1}; \lambda_1) + (\lambda_2 - \lambda_1)\|Z_{\lambda_1}\|_*$$
$$\iff \phi(Z_{\lambda_2}; \lambda_1) - \phi(Z_{\lambda_1}; \lambda_1) \leq (\lambda_2 - \lambda_1)(\|Z_{\lambda_1}\|_* - \|Z_{\lambda_2}\|_*)$$
$$\iff \phi(Z_{\lambda_2}; \lambda_1) - \phi(Z_{\lambda_1}; \lambda_1) \leq (\lambda_2 - \lambda_1)(\rho(\lambda_1) - \rho(\lambda_2)).$$

Notice $Z_{\lambda_1} \in \arg\min \phi(Z; \lambda_1)$, thus, we have $\phi(Z_{\lambda_2}; \lambda_1) - \phi(Z_{\lambda_1}; \lambda_1) \geq 0$, and it leads to $\rho(\lambda_1) \geq \rho(\lambda_2)$. Therefore, $\rho_\lambda$ is non-increasing in $\lambda$. Moreover, $\rho(\lambda_1) = \rho(\lambda_2)$ is achieved iff $\phi(Z; \lambda_1)$ and $\phi(Z; \lambda_2)$ admits a common solution and enjoys the same minimum objective value, i.e., $Z_{\lambda_1} = Z_{\lambda_2}$. When this happen, we study the optimality condition

$$\frac{\partial}{\partial Z}\phi(Z_{\lambda_1}; 0) + \lambda_1 \partial\|Z_{\lambda_1}\|_* = \frac{\partial}{\partial Z}\phi(Z_{\lambda_2}; 0) + \lambda_2 \partial\|Z_{\lambda_2}\|_* .$$

Therefore, one has

$$\partial\|Z_{\lambda_1}\|_* = \frac{\lambda_2}{\lambda_1}\partial\|Z_{\lambda_1}\|_* . \tag{32}$$

If $Z_{\lambda_1} = 0$ or $Z_{\lambda_2} = 0$, due to the condition that $\rho(\lambda_1) = \rho(\lambda_2)$, one must have $Z_{\lambda_1} = Z_{\lambda_2} = 0$. If they are both non-zero, due to the definition of subdifferential of nuclear norm, one has

$$\left\|\partial\|Z_{\lambda_1}\|_*\right\|_2 = \left\|\partial\|Z_{\lambda_2}\|_*\right\|_2 = 1 ,$$

which is contradictory to (32). Thus, when $\rho(\lambda_1) = \rho(\lambda_2)$ holds, one must have $Z_{\lambda_1} = Z_{\lambda_2} = 0$.

**Asymptotic behaviour of $\rho_\lambda$.** In this part, we aim to show

$$\lim_{\lambda \to 0} \rho_\lambda = \infty, \lim_{\lambda \to \infty} \rho_\lambda = 0 .$$

We first use proof by contradiction to show that $\lim_{\lambda \to \infty} \rho_\lambda = 0$. Assume there exists $M > 0$ such that $\forall \lambda > 0$, one has $\rho_\lambda \geq M$. Then, we take a series $\lambda_i$ such that $\lim_{i \to \infty} \lambda_i = \infty$, and let $S = \{\rho(\lambda_i)\}_{i=1}^\infty$. Based on the assumption, we know all the elements in $S$ has a lower bound $M$. Moreover, we know $\forall \lambda_i$, $\rho(\lambda_i) \leq \frac{\log K}{\lambda_i}$. This is because $\phi(0; \lambda) = \log K$ which is independent of the choices of $\lambda_i$. Thus, there exists $M'$ such that $\forall \rho_\lambda \in S$, we have $\rho_\lambda \leq M'$. Based on Bolzano–Weierstrass theorem, there exists a subsequence $\lambda_{i_k}$ such that $\lim_{k \to \infty} \rho(\lambda_{i_k}) = M^*$ where $M \leq M^* \leq M'$. However, in this case, $\lim_{k \to \infty} \phi(Z_{\lambda_{i_k}}^*; \lambda_{i_k}) \geq \lim_{k \to \infty} \rho(\lambda_{i_k})\lambda_{i_k}\rho(\lambda_{i_k}) = \infty$. This is in contradictory to the assumption that $Z_{\lambda_{i_k}}^*$ minimizes $\phi(Z; \lambda_{i_k})$ and $\phi(0; \lambda) = \log K$. Thus, one must have $\lim_{\lambda \to \infty} \rho_\lambda = 0$.

Next, we first use proof by contradiction to show that $\lim_{\lambda \to 0} \rho_\lambda = \infty$. Assume there exists $N > 0$ such that $\forall \lambda > 0$, one has $\rho_\lambda \leq N$. Let $L^* = \min_Z \phi(Z; 0)$. Our first observation is that the minimum $L^*$ is achieved when the norm of $Z$ diverges. Moreover, for every $R > 0$, there exists a $\epsilon_R > 0$ such that

$$L_R^* \geq \epsilon_R$$
$$L_R^* := \min_Z \phi(Z; 0) \quad \text{s.t.} \|Z\|_* \leq R .$$

Then, we pick a series $\lambda_k = \frac{1}{k^2}, R_k = k$, and choose

$$\tilde{Z}_k = k \begin{pmatrix} \mathbf{1}_n^\top & \mathbf{0}_n^\top & \mathbf{0}_n^\top & \cdots, & \mathbf{0}_n^\top \\ \mathbf{0}_n^\top & \mathbf{1}_n^\top & \mathbf{0}_n^\top & \cdots & \mathbf{0}_n^\top \\ \mathbf{0}_n^\top & \mathbf{0}_n^\top & \cdots & & \mathbf{0}_n^\top & \mathbf{1}_n^\top \\ & & \mathbf{0}_{(K-mn) \times mn} & & \end{pmatrix}$$

Based on the optimality of $Z^*_{\lambda_k}$, one must have

$$\phi(Z^*_{\lambda_k}; \lambda_k) \leq \phi(\tilde{Z}_k; \lambda_k) = \log\big(1 + (\bar{K} - 1)\exp(-k)\big) + \frac{\bar{K}\sqrt{n}}{k}\,.$$

On the other hand,

$$\phi(Z^*_{\lambda_k}; \lambda_k) \geq \phi(Z^*_{\lambda_k}; 0) \geq \epsilon_N > 0\,.$$

Combine these two inequalities together, one has

$$\epsilon_N \leq \log\big(1 + (\bar{K} - 1)\exp(-k)\big) + \frac{\bar{K}\sqrt{n}}{k}\,.$$

However, one can choose $k$ sufficiently large that the above inequality breaks. Thus, there cannot exist a $N > 0$ such that $\rho_\lambda \leq N$ holds $\forall \lambda > 0$. Therefore, one must have $\lim_{\lambda \to 0} \rho_\lambda = \infty$. $\qquad\square$

### F.3  PROOF OF LEMMA F.1

*Proof.* Our starting point is the following lemma which is developed in Hong & Ling (2023).

**Lemma F.2** (Cross entropy loss is strongly convex in restricted direction). *Define*

$$\phi(Z) = \frac{1}{\bar{K}n} \sum_{c=1}^{\bar{K}} \sum_{j=1}^{n} \mathcal{L}_{\mathrm{CE}}\big(y_c, Ze_{(c-1)n+j}\big)\,,$$

*where $y_c$ is a one-hot vector with $c$-th index equals one. Then $\phi(Z)$ is strongly convex in the direction $\Delta_Z \in \mathbb{R}^{K \times \bar{K}n}$ that belongs to $\{\Delta_Z : 1_K^\top \Delta_Z = 0\}$.*

We refer the readers to Lemma 5.1 in Hong & Ling (2023) for the proof. Based on Lemma F.2, for any $Z^*$ that minimizes $\phi(Z; \lambda) := \phi(Z) + \lambda\|Z\|_*$. We first apply a decomposition of $Z^*$ as follows

$$Z^* = \frac{1}{K}1_K 1_K^\top Z^* + (I_K - \frac{1}{K}1_K 1_K^\top)Z^*\,.$$

For simplicity, let $P = I_K - \frac{1}{K}1_K 1_K^\top$ be the projection matrix onto the space orthogonal to $1_K$. Based on the property of cross entropy loss, $\frac{1}{K}1_K 1_K^\top Z^*$ does not affect the value of the cross entropy term, i.e., $\phi(Z^*) = \phi(PZ^*)$. Moreover, let the compact SVD of $Z^*$ be $Z^* = U_Z \Sigma_Z V_Z^\top$. We first consider the case when $K \geq \bar{K}n$. Based on Lemma C.3, one has

$$\begin{aligned}
\|PZ^*\|_* &= \max_{V \in \mathbb{R}^{\bar{K}n \times \bar{K}n}, V^\top V = I_{\bar{K}n}} \mathrm{Tr}(PZ^*V) \\
&= \max_{V \in \mathbb{R}^{\bar{K}n \times \bar{K}n}, V^\top V = I_{\bar{K}n}} \mathrm{Tr}(PU_Z \Sigma_Z V_Z^\top V) \\
&\leq \max_{V \in \mathbb{R}^{\bar{K}n \times \bar{K}n}, V^\top V = I_{\bar{K}n}} \sum_{i=1}^{\mathrm{rank}(Z^*)} \sigma_i(PU_Z)\sigma_i(\Sigma_Z V_Z^\top V) \\
&= \sum_{i=1}^{\mathrm{rank}(Z^*)} \sigma_i(PU_Z)\sigma_i(Z^*) \qquad V = [V_Z, V_{Z,\perp}] \\
&\leq \sum_{i=1}^{\mathrm{rank}(Z^*)} \sigma_i(Z^*) = \|Z^*\|_*\,,
\end{aligned}$$

where the last inequality holds because $P$ is a contraction map, and equality is achieved if and only if $\sigma_i(PU_Z) = \sigma_i(U_Z) = 1, \forall i \leq \mathrm{rank}(Z^*)$. On the other hand, $P$ is an orthogonal projection, and $\|PU_Z\|_F = \|U_Z\|_F$ if and only if $U_Z$ lies in the range of $P$, and it leads to

$$Z^* = PZ^*, \qquad 1_K^\top Z^* = 0\,.$$

For the case when $K \leq \bar{K}n$, the analysis is the same. Based on Lemma F.2, constrained on the space $1_K^\top Z = 0$, the problem is strongly convex, and there exists a unique solution. Moreover, we also show that the optimal solution must lies in $1_K^\top Z = 0$. Thus, the solution is unique. $\qquad\square$

# G  PROOF OF THEOREM 3.2

In this section, we present the full version of Theorem 3.2.

For convenience, we will first define the following notations. Let $W_{\mathrm{LoRA}}^\lambda = W_{\mathrm{pre}} + B_\lambda^* A_\lambda^*$, and define the following shorthand for margins as: $\gamma_{\mathrm{pre}} = \gamma(W_{\mathrm{pre}}; \mathcal{D}_{\mathrm{pre}})$, $\gamma_{\mathrm{ft},\lambda} = \gamma(B_\lambda^* A_\lambda^*; \mathcal{D}_{\mathrm{ft}})$, and $\gamma_\lambda = \gamma(W_{\mathrm{LoRA}}^\lambda; \mathcal{D}_{\mathrm{pre}} \cup \mathcal{D}_{\mathrm{ft}})$. Additionally, given an $\bar{K}$-class max-margin classifier $W_{\mathrm{ft}}^* \in \mathbb{R}^{\bar{K} \times d}$ on the fine-tuning data, let $\gamma_{\mathrm{ft}}^* = \gamma(([W_{\mathrm{ft}}^*; \mathbf{0}_{(K-\bar{K}) \times d}]); \mathcal{D}_{\mathrm{ft}})$.

With these definitions in place, we now present our main theorem.

**Theorem G.1.** *Adopt the setup of Theorem 3.1 and let $\Theta_\lambda := \|B_\lambda^* A_\lambda^*\|_F$, the normalized margins of $W_{\mathrm{LoRA}}^\lambda$ on the union of pre-training and fine-tuning data can be characterized uniformly as follows:*

$$\gamma(W_{\mathrm{LoRA}}^\lambda; \mathcal{D}_{\mathrm{pre}}) = \gamma_{\mathrm{pre}} \frac{\rho_{\mathrm{pre}}}{\sqrt{\Theta_\lambda^2 + \rho_{\mathrm{pre}}^2}}, \quad \gamma(W_{\mathrm{LoRA}}^\lambda; \mathcal{D}_{\mathrm{ft}}) = \gamma_{\mathrm{ft},\lambda} \frac{\Theta_\lambda}{\sqrt{\Theta_\lambda^2 + \rho_{\mathrm{pre}}^2}},$$

$$\gamma_\lambda = \min\left\{ \gamma(W_{\mathrm{LoRA}}^\lambda; \mathcal{D}_{\mathrm{pre}}), \gamma(W_{\mathrm{LoRA}}^\lambda; \mathcal{D}_{\mathrm{ft}}) \right\} \tag{33}$$

*Moreover, $\Theta_\lambda, \gamma_{\mathrm{ft},\lambda}$ takes different values in:*

(i) **High–penalty regime** $\lambda \geq \frac{1}{K\sqrt{n}}$: $\Theta_\lambda = \gamma_{\mathrm{ft},\lambda} = 0$.

(ii) **Intermediate regime** $\lambda_{\mathrm{crit}} < \lambda < \frac{1}{K\sqrt{n}}$: $\Theta_\lambda = \frac{\rho_\lambda}{\sqrt{\bar{K}-1}}$ *and* $\gamma_{\mathrm{ft},\lambda} = \gamma_{\mathrm{ft}}^*$. *Moreover, $\Theta_\lambda$ is a strictly decreasing function w.r.t. $\lambda$.*

(iii) **Low–penalty regime** $\gamma_{\mathrm{ft},\lambda} < \gamma_{\mathrm{ft}}^*$ *and*

$$\Theta_\lambda := \sqrt{n\bar{K}a_\lambda^2 + n\bar{K}(\bar{K}-1)b_\lambda^2 + n\bar{K}(K-\bar{K})c_\lambda^2}, \quad \gamma_{\mathrm{ft},\lambda} = \frac{a_\lambda + c_\lambda}{\Theta_\lambda}.$$

**Optimal trade-off choice of $\lambda$.** *Assume $\rho_{\mathrm{pre}} \leq \frac{\Theta_{\lambda_{\mathrm{crit}}} \gamma_{\mathrm{ft}}^*}{\gamma_{\mathrm{pre}}}$, then there exists a unique $\lambda^*$ such that*

$$\max_{\lambda_{\mathrm{crit}} < \lambda < \frac{1}{K\sqrt{n}}} \gamma_\lambda = \frac{\gamma_{\mathrm{ft}}^* \gamma_{\mathrm{pre}}}{\sqrt{(\gamma_{\mathrm{ft}}^*)^2 + \gamma_{\mathrm{pre}}^2}}, \qquad \text{attained at } \lambda = \lambda^*. \tag{34}$$

*Proof.* Our key observation is that when one computes margin of $W_{\mathrm{LoRA}}^\lambda$ on the pre-training and fine-tuning dataset, only $W_{\mathrm{pre}}$ or $B_\lambda^* A_\lambda^*$ will be activated. This is because due to Theorem 3.1, we have shown that the product of the optimal LoRA adapters lies in the span of fine-tuning data, and due to Proposition C.1 and Assumption, one also can show that $W_{\mathrm{pre}}$ lies in the span of pre-training data. Thus, under Assumption 2.3, we can conclude $B_\lambda^* A_\lambda^* x = 0$ if $x \in \mathcal{D}_{\mathrm{pre}}$, and $W_{\mathrm{pre}} x = 0$ if $x \in \mathcal{D}_{\mathrm{ft}}$. Before beginning prove the margin takes the structures as is shown in Theorem 3.2, we first introduce the following notations. Let $W_{\mathrm{pre}} = [w_1, \ldots, w_K]^\top$, $B_\lambda^* A_\lambda^* = [\delta w_{\lambda,1}, \cdots, \delta w_{\lambda,K}]^\top$.

Now, when we compute the margin of $W_{\mathrm{LoRA}}^\lambda$ on pre-training data, one has

$$\gamma(W_{\mathrm{LoRA}}^\lambda; \mathcal{D}_{\mathrm{pre}}) = \frac{\min_{(x_{i,c}, y_c) \in \mathcal{D}_{\mathrm{pre}}} (w_c + \delta_{\lambda,c})^\top x - \max_{i \neq y}(w_i + \delta_{\lambda,i})^\top x}{\|W_{\mathrm{LoRA}}^\lambda\|_F}$$

$$= \frac{\min_{(x_{i,c}, y_c) \in \mathcal{D}_{\mathrm{pre}}} w_c^\top x - \max_{i \neq y} w_i^\top x}{\|W_{\mathrm{pre}}\|_F} \cdot \frac{\|W_{\mathrm{pre}}\|_F}{\|W_{\mathrm{LoRA}}^\lambda\|_F}$$

$$= \gamma_{\mathrm{pre}} \cdot \frac{\rho_{\mathrm{pre}}}{\sqrt{\rho_{\mathrm{pre}}^2 + \|B_\lambda^* A_\lambda^*\|_F^2}}.$$

Then, for the fine-tuning data, one has

$$
\gamma(W_{\text{LoRA}}^\lambda; \mathcal{D}_{\text{ft}}) = \frac{\min_{(x_{i,c}, y_c) \in \mathcal{D}_{\text{ft}}} (w_c + \delta_{\lambda,c})^\top x - \max_{i \neq y}(w_i + \delta_{\lambda,i})^\top x}{\|W_{\text{LoRA}}^\lambda\|_F}
$$

$$
= \frac{\min_{(x_{i,c}, y_c) \in \mathcal{D}_{\text{ft}}} \delta_{\lambda,c}^\top x - \max_{i \neq y} \delta_{\lambda,i}^\top x}{\|B_\lambda^* A_\lambda^*\|_F} \cdot \frac{\|B_\lambda^* A_\lambda^*\|_F}{\sqrt{\rho_{\text{pre}}^2 + \|B_\lambda^* A_\lambda^*\|_F^2}},
$$

$$
= \gamma_{\text{ft},\lambda} \cdot \frac{\|B_\lambda^* A_\lambda^*\|_F}{\sqrt{\rho_{\text{pre}}^2 + \|B_\lambda^* A_\lambda^*\|_F^2}}
$$

It suffices to compute $\|B_\lambda^* A_\lambda^*\|_F$ for different regimes of the regularization level.

**High-penalty regime.** In this regime, $B_\lambda^* A_\lambda^* = 0$, so one has $\|B_\lambda^* A_\lambda^*\|_F = 0$.

**Intermediate regime.** In the intermediate regime, one has

$$
B_\lambda^* A_\lambda^* = \frac{\rho_\lambda}{\sqrt{\bar{K} n (\bar{K} - 1)}} \left( \begin{pmatrix} M_{\bar{K}} \\ \mathbf{0}_{(K-\bar{K}) \times \bar{K}} \end{pmatrix} \otimes \mathbf{1}_n^\top \right) X_{\text{ft}}^\top , \tag{35}
$$

and one can compute its norm as follows

$$
\|B_\lambda^* A_\lambda^*\|_F^2 = \frac{\rho_\lambda^2}{mn(m-1)} \cdot \frac{\bar{K}}{\bar{K}-1} \left\| \left( \left( \mathbf{I}_{\bar{K}} - \frac{1}{\bar{K}} \mathbf{1}_{\bar{K}} \mathbf{1}_{\bar{K}}^\top \right) \right) \otimes \mathbf{1}_n^\top X_{\text{ft}}^\top \right\|_F^2
$$

$$
= \frac{\rho_\lambda^2}{\bar{K} n (\bar{K}-1)} \cdot \frac{\bar{K}}{\bar{K}-1} \cdot \bar{K} \left( n \cdot (\frac{\bar{K}-1}{\bar{K}})^2 + \frac{(\bar{K}-1)n}{\bar{K}^2} \right)
$$

$$
= \frac{\rho_\lambda^2}{\bar{K}-1} .
$$

Thus, $\|B_\lambda^* A_\lambda^*\|_F = \|\bar{B}_\lambda^* \bar{A}_\lambda^*\|_F^2 = \frac{\rho_\lambda}{\sqrt{\bar{K}-1}}$.

**Low-penalty regime.** In this regime, $B_\lambda^* A_\lambda^*$ takes the following form

$$
B_\lambda^* A_\lambda^* = \left( \begin{pmatrix} (a_\lambda + b_\lambda) \mathbf{I}_{\bar{K}} - b_\lambda \mathbf{1}_{\bar{K}} \mathbf{1}_{\bar{K}}^\top \\ -c_\lambda \mathbf{1}_{K-\bar{K}} \mathbf{1}_{\bar{K}}^\top \end{pmatrix} \otimes \mathbf{1}_n^\top \right) X_{\text{ft}}^\top , \tag{36}
$$

and Frobenius norm is

$$
\|\bar{B}_\lambda^* \bar{A}_\lambda^*\|_F^2 = \sqrt{n} \left\| (a_\lambda + b_\lambda) \mathbf{I}_{\bar{K}} - b_\lambda \mathbf{1}_{\bar{K}} \mathbf{1}_{\bar{K}}^\top \right\|_F
$$

$$
= \sqrt{n} \cdot \sqrt{\bar{K} a_\lambda^2 + \bar{K}(\bar{K}-1) b_\lambda^2} ,
$$

and

$$
\|B_\lambda^* A_\lambda^*\|_F = \sqrt{n} \left\| \begin{pmatrix} (a_\lambda + b_\lambda) \mathbf{I}_{\bar{K}} - b_\lambda \mathbf{1}_{\bar{K}} \mathbf{1}_{\bar{K}}^\top \\ -c_\lambda \mathbf{1}_{K-\bar{K}} \mathbf{1}_{\bar{K}}^\top \end{pmatrix} \right\|_F
$$

$$
= \sqrt{n} \cdot \sqrt{\bar{K} a_\lambda^2 + \bar{K}(\bar{K}-1) b_\lambda^2 + (K-\bar{K})\bar{K} c_\lambda^2} .
$$

Finally, we study the optimal choices of $\lambda$. First, it is obvious that $\gamma(W_{\text{LoRA}}^\lambda; \mathcal{D}_{\text{pre}})$ is a decreasing function w.r.t. $\bar{\Theta}_\lambda$ and $\gamma(W_{\text{LoRA}}^\lambda; \mathcal{D}_{\text{ft}})$ is an increasing function w.r.t. $\Theta_\lambda$. In the *intermediate regime* where $\Theta_\lambda = \bar{\Theta}_\lambda$, the optimal $\rho_\lambda$ is achieved when

$$
\gamma(W_{\text{LoRA}}^\lambda; \mathcal{D}_{\text{pre}}) = \gamma(W_{\text{LoRA}}^\lambda; \mathcal{D}_{\text{ft}}) .
$$

In Appendix E, we have shown that $\rho_\lambda$ is a continuous decreasing function w.r.t. $\lambda$. Thus, $\rho_{\text{pre}} \leq \frac{\rho(\lambda_{\text{crit}}) \gamma_{\text{ft}}}{\sqrt{\bar{K}-1} \gamma_{\text{pre}}}$ implies $\rho^* := \frac{\sqrt{\bar{K}-1} \rho_{\text{pre}} \gamma_{\text{pre}}}{\gamma_{\text{ft}}} \leq \rho(\lambda_{\text{crit}})$ which further implies the corresponding regularization level $\lambda^*$ lies in the *intermediate regime*. Moreover, one can compute that in this case,

$$
\gamma(W_{\text{LoRA}}^{\lambda^*}; \mathcal{D}_{\text{pre}}) = \gamma(W_{\text{LoRA}}^{\lambda^*}; \mathcal{D}_{\text{ft}}) = \frac{\gamma_{\text{pre}} \gamma_{\text{ft}}^*}{\gamma_{\text{pre}}^2 + (\gamma_{\text{ft}}^*)^2} .
$$

Furthermore, we have shown that $\rho^*$ must be the solution of (28) in Appendix E. We can simply plug in the expression of $\rho^*$ to get the corresponding optimal $\lambda^*$.

$\square$

## H   PROOF OF THEOREM 3.3

*Proof.* Following the same argument as in Appendix G, one can show that the margin of linear classifier $W_{\text{LoRA}}$ for each task is

$$\gamma\big(W_{\text{LoRA}}(\boldsymbol{\alpha}); \mathcal{D}_{\text{pre}}\big) = \gamma_{\text{pre}} \frac{\rho_{\text{pre}}}{\sqrt{\rho_{\text{pre}}^2 + \sum_{j=1}^{T} \alpha_j^2 \Theta_{\lambda,j}^2}}, \tag{37}$$

$$\gamma\big(W_{\text{LoRA}}(\boldsymbol{\alpha}); \mathcal{D}_i\big) = \gamma_i \frac{\alpha_i \Theta_{\lambda,j}}{\sqrt{\rho_{\text{pre}}^2 + \sum_{j=1}^{T} \alpha_j^2 \Theta_{\lambda,j}}}, \quad i = 1, \ldots, T. \tag{38}$$

Then, our goal is to solve the following optimization problem

$$\max_{\alpha_1, \cdots, \alpha_T} \min\left(\gamma_{\text{pre}} \frac{\rho_{\text{pre}}}{\sqrt{\rho_{\text{pre}}^2 + \sum_{j=1}^{T} \alpha_j^2 \Theta_{\lambda,j}^2}}, \quad \min_{i \leq T} \gamma_i \frac{\alpha_i \Theta_{\lambda,j}}{\sqrt{\rho_{\text{pre}}^2 + \sum_{j=1}^{T} \alpha_j^2 \Theta_{\lambda,j}}}\right)$$

For convenience, we introduce the following notation $x_i = \alpha_i \Theta_{\lambda,i}, i \in [T]$, then the above optimization problem can be rewritten as

$$\max_{x_1, \cdots, x_T} \min\left(\frac{\rho_{\text{pre}}\gamma_{\text{pre}}}{\sqrt{\rho_{\text{pre}}^2 + \sum_{j=1}^{T} x_j^2}}, \quad \min_{i \in [T} \frac{\gamma_i x_n}{\sqrt{\rho_{\text{pre}}^2 + \sum_{j=1}^{T} x_j^2}}\right). \tag{39}$$

Let $S := \sqrt{\rho_{\text{pre}}^2 + \sum_{j=1}^{T} x_j^2}$ and $t := \min\{\frac{\gamma_i x_i}{S}, \frac{\gamma_{\text{pre}}\rho_{\text{pre}}}{S}\}$, then Problem 39 is equivalent to

$$\max_{t,S,x_1,\cdots,x_T} t \tag{40}$$

$$\text{s.t.} \quad \gamma_i x_i \geq tS, i \in [T]$$

$$\gamma_{\text{pre}}\rho_{\text{pre}} \geq tS,$$

$$\sum_{i=1}^{T} x_i^2 + \rho_{\text{pre}}^2 = S^2, \quad S > 0.$$

We first claim at optimum, the inequalities w.r.t. $x_i$ will be tight, i.e., $x_i = \frac{tS}{\gamma_i}, \forall i \in [T]$. In this case, one can see that

$$S^2 = \rho_{\text{pre}}^2 + \sum_{j=1}^{T} x_j^2 \geq S^2 t^2 \cdot \left(\sum_{i=1}^{T} \frac{1}{\gamma_i^2} + \frac{1}{\gamma_{\text{pre}}^2}\right).$$

Thus, the optimal value of the objective is $t \leq \frac{1}{\sqrt{\frac{1}{\gamma_{\text{pre}}^2} + \sum_{j=1}^{T} \frac{1}{\gamma_j^2}}}$, and the equality is achieved under the condition $\gamma_i x_i = \gamma_{\text{pre}}\rho_{\text{pre}}$, which is equivalent to $\alpha_i = \frac{\rho_{\text{pre}}\gamma_{\text{pre}}}{\gamma_i \Theta_{\lambda,i}}$, $i = 1, \ldots, T$.

Finally, we show why the optimum of Problem 40 is achieved when all inequalities w.r.t. $x_i$ become equality.

**Suppose one inequality is slack.** Assume for the optimal solution $(t^*, S^*, \boldsymbol{x}^*)$ where $\boldsymbol{x}^* = (x_1^*, \cdots, x_T^*)$, there exists $j \in [T]$ such that $r_j x_j^* > t^* S^*$.

**Shrink $x_j^*$ while fix $t^*$.** As we decrease $x_j^*$ slightly, i.e., $x_j^* \to x_j^* - \epsilon, \epsilon > 0$. Then, the new $S$ will be

$$\tilde{S} = \sqrt{\rho_{\text{pre}}^2 + \sum_{i \neq j}(x_i^*)^2 + ((x_j^*) - \epsilon)^2} < S^*$$

When $\epsilon$ is sufficiently small, all the inequality w.r.t. other index will be slack since $\tilde{S} < S$, and $\gamma_i(x_j^* - \epsilon) > t^* \tilde{S}$. Therefore, one can safely increase $t^*$ a little until one of the inequality becomes tight, and it leads a larger objective value. Therefore, none of the inequality can be slack at global optimum. $\square$

# I  EXPERIMENT

In this section, we present the detailed setup of the experiments in §4.

**Definition of superclasses of CIFAR-100.** CIFAR-100 groups its 100 fine categories into **20 coarse** *superclasses* as follows

Table 1: CIFAR-100 coarse superclasses and their five fine labels.

| Superclass | Fine classes (5 per superclass) |
|---|---|
| aquatic mammals | beaver, dolphin, otter, seal, whale |
| fish | aquarium fish, flatfish, ray, shark, trout |
| flowers | orchid, poppy, rose, sunflower, tulip |
| food containers | bottle, bowl, can, cup, plate |
| fruit & vegetables | apple, mushroom, orange, pear, sweet pepper |
| household electrical devices | clock, computer keyboard, lamp, telephone, television |
| household furniture | bed, chair, couch, table, wardrobe |
| insects | bee, beetle, butterfly, caterpillar, cockroach |
| large carnivores | bear, leopard, lion, tiger, wolf |
| large man-made outdoor things | bridge, castle, house, road, skyscraper |
| large natural outdoor scenes | cloud, forest, mountain, plain, sea |
| large omnivores & herbivores | camel, cattle, chimpanzee, elephant, kangaroo |
| medium-sized mammals | fox, porcupine, possum, raccoon, skunk |
| non-insect invertebrates | crab, lobster, snail, spider, worm |
| people | baby, boy, girl, man, woman |
| reptiles | crocodile, dinosaur, lizard, snake, turtle |
| small mammals | hamster, mouse, rabbit, shrew, squirrel |
| trees | maple, oak, palm, pine, willow |
| vehicles 1 | bicycle, bus, motorcycle, pickup truck, train |
| vehicles 2 | lawn-mower, rocket, streetcar, tank, tractor |

**Construction of pre-training and fine-tuning tasks.** For every superclass $S$, let its ordered fine labels be $[c_1^S, \ldots, c_5^S]$. We build three disjoint labelled sets

$$\mathcal{D}_{\text{pre}} = \bigcup_S \{c_1^S, c_2^S, c_3^S\}, \quad \mathcal{D}_1 = \bigcup_S \{c_4^S\}, \quad \mathcal{D}_2 = \bigcup_S \{c_5^S\}.$$

Hence $\mathcal{D}_{\text{pre}}$ contains 60 fine classes (three per super-class) while each fine-tuning task $\mathcal{D}_i$ contains exactly one *new* class per superclass, preserving maximal diversity yet zero overlap with $\mathcal{D}_{\text{pre}}$.

**Frozen feature extractors.**  We evaluate four widely used backbones, noting their different pre-training dataset:

- **ResNet-50** (He et al., 2016) (torchvision, *supervised ImageNet-1K*).

- **ViT-Base/16** (Dosovitskiy et al., 2021) (timm: first self-supervised on *ImageNet-21K*, then fine-tuned on ImageNet-1K).

- **ConvNeXt-Tiny** (Liu et al., 2022) (timm: identical 21K → 1K pipeline as ViT).

- **CLIP ViT-B/32** (Radford et al., 2021) (OpenAI's contrastive pre-training on *web-scale image–text* pairs; no ImageNet supervision).

All models are kept frozen; we extract the CLS token (ViT/CLIP) or the global-average-pooled penultimate tensor (CNNs) at $224 \times 224$ resolution for every CIFAR-100 image.

**Hardware.** All runs were executed on a single NVIDIA RTX A5000 (24 GB). End-to-end required less than 1 hour of wall-clock time.

Figure 3: Distributions of pairwise feature correlation across four pre-trained models. The red dashed line represent the mean value of correlations, and the dashed lines represent mean plus and minus one std.

**Pre-training stage.** A linear classifier $W_{\text{pre}} \in \mathbb{R}^{20 \times d}$ ($d$ = embedding dimension) is trained from scratch on $\mathcal{D}_{\text{pre}}$ for 2000 epochs with Adam ($\eta = 0.1$). The 20 rows correspond to the super-classes, not the 60 fine labels; this matches our theoretical model where each task's labels share a common output.

### I.1 Validation of Orthogonal Data Assumption

In this section, we conduct a comprehensive feature correlation analysis on CIFAR-100. This analysis quantifies both intra-class and inter-class correlations in the feature space to numerically validate Assumption 2.3.

Let $\mathcal{F}_\theta(x)$ denote the feature extraction function of a neural network with parameters $\theta$, where $x \in \mathbb{R}^{3 \times 32 \times 32}$ represents an input image from CIFAR-100. For each architecture (ResNet-50, ViT-Base/16, CLIP ViT-B/32, and ConvNeXt-Tiny), we extract features from the penultimate layer, obtaining feature vectors $f_i = \mathcal{F}_\theta(x_i) \in \mathbb{R}^d$ for each training sample $x_i$.

We organize the extracted features by class, creating sets $\mathcal{S}_c = \{f_i : y_i = c\}$ for each class $c \in \{0, 1, \ldots, 99\}$, where $y_i$ is the ground truth label of sample $x_i$.

**Correlation Computation.** For any two feature vectors $f_i, f_j \in \mathbb{R}^d$, we compute the Pearson correlation coefficient:

$$\rho(f_i, f_j) = \frac{\text{Cov}(f_i, f_j)}{\sigma(f_i)\sigma(f_j)} = \frac{\sum_{k=1}^d (f_i^{(k)} - \bar{f}_i)(f_j^{(k)} - \bar{f}_j)}{\sqrt{\sum_{k=1}^d (f_i^{(k)} - \bar{f}_i)^2}\sqrt{\sum_{k=1}^d (f_j^{(k)} - \bar{f}_j)^2}} \tag{41}$$

| | CLIP | RESNET | VIT | CONVNEXT |
|---|---|---|---|---|
| Intra-class mean (std) | 0.2725(0.0797) | 0.2303(0.0701) | 0.2535(0.0708) | 0.2739(0.0820) |
| Inter-class mean (std) | -0.0022(0.0679) | -0.0024(0.0630) | -0.0022(0.0573) | -0.0026(0.0447) |

Table 2: Mean and standard deviation of intra-class and inter-class feature correlations. Values closer to zero indicate greater orthogonality between features.

where $f_i^{(k)}$ denotes the $k$-th dimension of feature vector $f_i$, and $\bar{f}_i = \frac{1}{d} \sum_{k=1}^{d} f_i^{(k)}$ is the mean of $f_i$.

**Intra-class Correlation.** For each class $c$, we compute the average intra-class correlation:

$$\rho_{\text{intra}}(c) = \frac{1}{|\mathcal{S}_c|(|\mathcal{S}_c| - 1)/2} \sum_{f_i, f_j \in \mathcal{S}_c, i < j} \rho(f_i, f_j) \tag{42}$$

This measures how similar features are within the same class, indicating the consistency of learned representations for semantically similar samples.

**Inter-class Correlation.** For any two distinct classes $c_1 \neq c_2$, we compute the average inter-class correlation:

$$\rho_{\text{inter}}(c_1, c_2) = \frac{1}{|\mathcal{S}_{c_1}| \cdot |\mathcal{S}_{c_2}|} \sum_{f_i \in \mathcal{S}_{c_1}} \sum_{f_j \in \mathcal{S}_{c_2}} \rho(f_i, f_j) \tag{43}$$

This quantifies the similarity between features from different classes, with lower values indicating better class separation.

**Correlation Matrix Construction.** We construct a $100 \times 100$ correlation matrix $\mathbf{R}$ where:

$$R_{ij} = \begin{cases} \rho_{\text{intra}}(i) & \text{if } i = j \\ \rho_{\text{inter}}(i, j) & \text{if } i \neq j \end{cases} \tag{44}$$

**Implementation Details.** To ensure computational efficiency while maintaining statistical reliability, we subsample feature pairs for correlation computation. Specifically, for each class we randomly sample 100 points (100,000 in total). We first report the distribution of pairwise correlations across all sampled points in Figure 3. To further distinguish intra-class and inter-class relationships, we visualize the corresponding correlations as a heatmap in Figure 4. For intra-class correlations, we randomly sample up to 100 pairs per class when the number of possible pairs exceeds this threshold; for inter-class correlations, we sample up to 100 pairs for each class pair. This subsampling strategy yields robust correlation estimates while keeping the computation tractable.

Figure 3 highlights two key observations: (1) most correlations are concentrated around zero, supporting Assumption 2.3 of orthogonal features; and (2) the mean correlation $\pm$ one standard deviation remains below 0.14, indicating that although the features are not perfectly orthogonal, their pairwise angles are sufficiently close to orthogonal for our analysis.

Figure 4 illustrates three key observations: (1) higher diagonal values indicate consistent within-class representations; (2) lower off-diagonal values suggest strong class separation; and (3) the empirical mean correlations for intra-class inter-class, indicate that features across classes are nearly orthogonal while within-class features exhibit only moderate correlation (See Table 2). These findings provide empirical support for Assumption 2.3, which posits orthogonality across both inter-class and intra-class features. Extending the analysis to settings with more complex cluster structure or stronger intra-class correlations remains an important direction for future work.

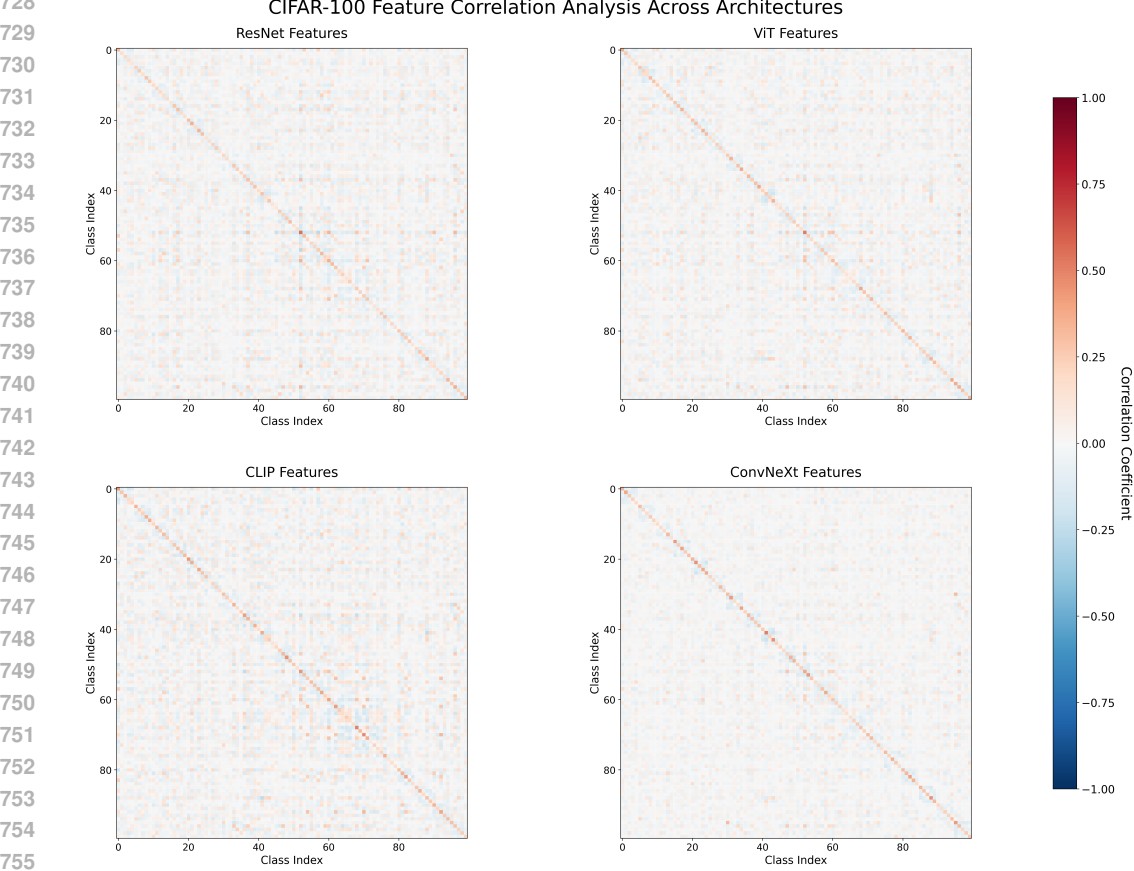

Figure 4: Feature correlation analysis across different pretrained models.

## I.2 DETAILS FOR EXPERIMENTS IN §4.1

For each fine-tuning task, we freeze the pre-trained weights $W_{\text{pre}}$ and attach a rank-$r$ LoRA block, where $r$ is set to the number of classes in the fine-tuning task. This satisfies Assumption 2.2. We evaluate performance across three fine-tuning tasks with varying class counts: $\bar{K} = 5, 10, 20$. Each task is constructed by selecting the top $\bar{K}$ superclasses from $\mathcal{D}_2$.

We fine-tune the model on each task using SGD for 3000 steps with a learning rate of 0.5, sweeping over regularization values $\lambda \in [10^{-5}, 10^{-1}]$ with 50 logarithmically spaced regularization strengths. For each $\bar{K}$ and each frozen feature extractor, we record both the empirically optimal regularization parameter and the theoretically predicted one from Theorem 3.2. The detailed figures are given as follows:

## I.3 DETAILS FOR EXPERIMENTS IN §4.2

For the experiments in §4.2, we fine-tune $\mathcal{D}_1, \mathcal{D}_2$ using different LoRA adapters $(B_i, A_i), i = 1, 2$ with LoRA rank 20. We fine-tune the LoRA adapters using Adam ($\eta = 0.1$) with regularization parameter $\lambda = 5 \times 10^{-7}$. Let $(B_i^*, A_i^*), i = 1, 2$ be the optimal LoRA adapters we achieve at the end of fine-tuning, we merge them together as follows

$$W_{\text{LoRA}}^\lambda(\alpha_1, \alpha_2) = W_{\text{pre}} + \alpha_1 B_1^* A_1^* + \alpha_2 B_2^* A_2^*$$

To seek for the optimal mixing coefficients $(\alpha_1, \alpha_2)$, we run grid search over a $50 \times 50$ lattice on $(0, 1)^2$, and compare it with the optimal theoretical mixing coefficients presented in Theorem 3.3. We test different number of classes in the fine-tuning datasets, i.e., $\bar{K} = 5, 10, 20$, and reports their results here.

Table 3: Empirical (Emp) and theoretical (Thm) optimal regularization parameters $\lambda$ for different pre-trained models and fine-tuning task sizes ($\bar{K}$).

| Model | $\bar{K} = 5$ | $\bar{K} = 10$ | $\bar{K} = 20$ |
|---|---|---|---|
| ResNet-50 | Emp: 0.002189 | Emp: 0.002189 | Emp: 0.001092 |
| | Thm: 0.007424 | Thm: 0.004158 | Thm: 0.002121 |
| ViT-B/16 | Emp: 0.000918 | Emp: 0.001299 | Emp: 0.000648 |
| | Thm: 0.003304 | Thm: 0.002763 | Thm: 0.001551 |
| ConvNeXt | Emp: 0.000458 | Emp: 0.001299 | Emp: 0.000648 |
| | Thm: 0.006709 | Thm: 0.003114 | Thm: 0.002039 |
| CLIP | Emp: 0.003687 | Emp: 0.001839 | Emp: 0.000771 |
| | Thm: 0.001904 | Thm: 0.001185 | Thm: 0.001032 |

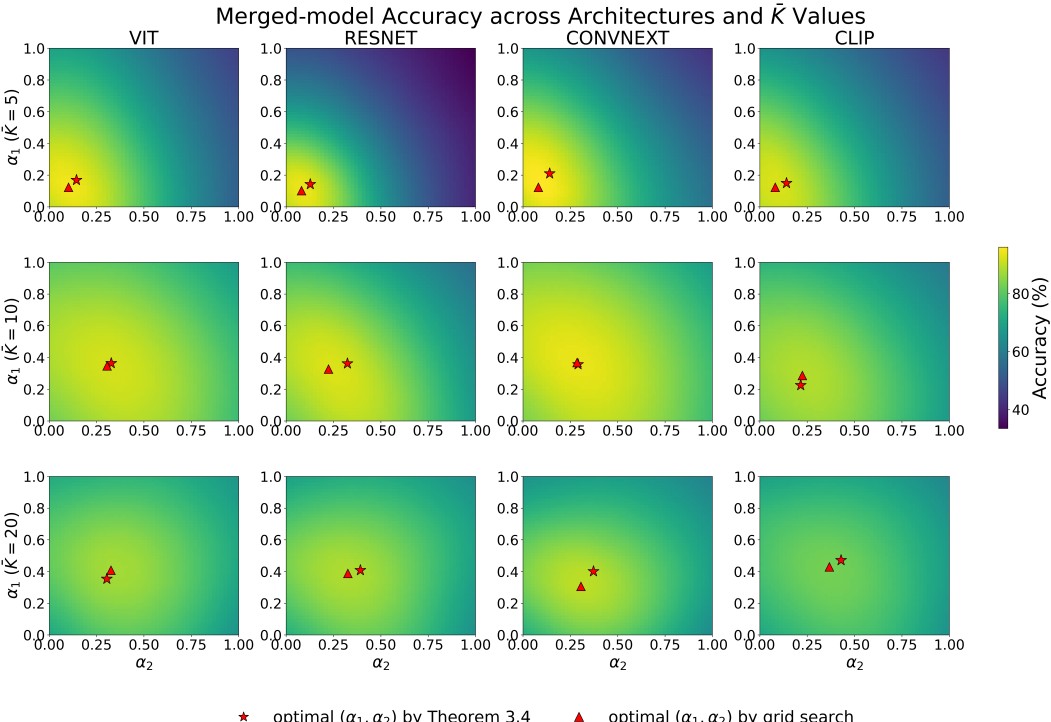

Figure 5: Merged-model accuracy across architectures and different number of classes in the fine-tuning tasks. Each panel shows the accuracy of the merged model evaluated on the combined dataset, across a $50 \times 50$ grid of mixing coefficients $(\alpha_1, \alpha_2) \in (0, 1)^2$. For each architecture, the red star indicates the theoretically predicted optimal coefficients from Theorem 3.3, while the red triangle marks the empirically optimal coefficients.

