# OpenReview forum: "LoRA Provably Reduces Forgetting and Enables Adapter Merging in Multiclass Linear Classification"
_ICLR.cc/2026/Conference — Submitted to ICLR 2026_

### Official Review · Reviewer_85Bk · 2025-10-27

**Soundness:** 2
**Presentation:** 2
**Contribution:** 2
**Rating:** 4
**Confidence:** 3

**Summary:**

This paper provides a theoretical analysis of the Low-Rank Adaptation (LoRA) algorithm in the context of multiclass linear classification. The authors aim to offer a principled explanation for two important empirical findings: (i) LoRA's resistance to catastrophic forgetting, and (ii) the ability to merge independently trained adapters into a single model that performs well on multiple tasks. The paper builds on the concept of maximizing the margin in linear classification and derives closed-form expressions for optimal LoRA adapters, showing that LoRA’s effectiveness in forgetting reduction and adapter merging can be understood through the lens of margin theory.

**Strengths:**

1. The paper has a good the high-level motivation, i.e., to explain LoRA’s empirical behavior from first principles, which is timely and relevant to the community.
2. Given the strong simplifying assumptions, the theoretical derivations are internally consistent. The link between the regularized LoRA objective and hard-margin SVM theory is clearly explained.

**Weaknesses:**

1. The Assumption 2.3 is a strong and unrealistic assumption in practical scenarios. The author provide the some numerical evidence in Appendix I.1 by using the pearson correlation coefficient, it may not be sufficient to support such a restrictive orthogonal assumption. Same for the Assumption 3.1.
2. This paper only consider the multiclass linear classification task, but the usage of LoRA are often focus on the large scale model such as Large Language Models on the generation tasks.
3. The experiments are conducted on LoRA with only the one layer of the model (final linear), can the result expand to multiple layer as many exsiting usage of LoRA?

**Questions:**

1. This paper are based on the ''orthogonal tasks'', which lacks of the detailed explaination. Are the orthogonal tasks refers to the task that satisfied the Assumptions 2.3? If that, the reference that ''across all original tasks'' in the introduction (line 58) seems problematic as many of them do not have this assumption.
2. While the paper provides a deep theoretical analysis, it could benefit from a more detailed discussion of the practical implications of LoRA's findings for real-world applications, such as incremental learning or continual learning. How do the theoretical insights apply to large-scale models like LLMs (Large Language Models)?
3. How performance changes as correlation of task increases?

---

> ### Author Response · Authors · 2025-11-26
>
> We thank the reviewers for the comments. Please see the general responses where we address your concerns raised in **Weakness** and **Q1**.
>
> **Response to Q2** We thank the reviewer for suggesting a discussion on incremental learning and continual learning. These directions are indeed related and can be included as potential extensions of our current results in the future work section.
>
> **Response to Q3** In this work, we assume that tasks are orthogonal to each other, which can be intuitively interpreted as having zero task correlation. While we currently do not have theoretical guarantees for the case where pre-training and fine-tuning tasks are positively correlated, our empirical observations indicate that the theoretical insights continue to hold in this regime. In particular, we find that the learned LoRA adapters still align closely with the max-margin direction of the fine-tuning data.

---

### Official Review · Reviewer_Qs8n · 2025-10-29

**Soundness:** 3
**Presentation:** 3
**Contribution:** 1
**Rating:** 2
**Confidence:** 4

**Summary:**

The authors present a theoretical study on how LoRA exhibits resistance to catastrophic forgetting and enables the effective merging of separately fine-tuned models. Starting from a set of simplifying assumptions and leveraging the theory of soft and hard margins in SVMs, they develop formal theorems and proofs showing that, when models are fine-tuned with LoRA under norm regularization on the LoRA matrices, three distinct training regimes emerge, each leading to different model behaviors. The analysis further demonstrates that LoRA imposes specific boundaries between pretraining and finetuning phases depending on the strength of the regularization term, and that optimal scaling factors exist which *i)* allow the resulting model to find the most effective boundary between pretraining and finetuning tasks, thus mitigating catastrophic forgetting, or *ii)* yield optimal merging performance across multiple fine-tuned models. Theoretical insights are supported by experiments on various vision architectures using the CIFAR-100 dataset.

**Strengths:**

- Mathematical completeness: assumptions, lemmas and theorems are clearly stated, and proofs are provided
- Overall, the paper is well written, the notation is precise and not too complex at the same time

**Weaknesses:**

1. Experimental section: in my opinion, this is the main weakness of the paper. The authors make several simplifying assumptions to obtain more tractable mathematics (see next weakness), which, while potentially too restrictive and simplistic, could be acceptable if the experimental section demonstrated that the theoretical insights generalize to more realistic and complex settings. However, this is not the case. To validate their findings, the authors conduct linear probing experiments: *i.e.*, they fine-tune only the final classification layer with LoRA while keeping the rest of the model frozen. This setup is both unrealistic, since LoRA is typically applied to fine-tune the backbone (particularly the Attention modules) rather than the classifier head, and overly simplistic. Consequently, the empirical results have limited practical relevance. Moreover, given that the stated goal of the paper is to explain how and why LoRA mitigates catastrophic forgetting and facilitates model merging, the comparison baseline should include Full Fine-Tuning (FFT). The authors briefly mention FFT in Section 3.2, noting that comparing LoRA to FFT is not their goal: this, however, appears inconsistent with the paper’s central motivation.


2. Strong assumptions, which may rarely hold in practice:
   - Assumption 2.1, *i.e.* "The input data dimension is larger than the total number of classes", is reasonable.
   - Assumption 2.2, while somewhat restrictive, is still acceptable if it helps maintain mathematical tractability: LoRA can and has been used on datasets with more classes than LoRA’s rank.
   - Assumptions 2.3 and 2.4 are overly restrictive, and lack justification that they are met in realistic scenarios. Even Section I.1 (page 31 of the Appendix), which attempts to validate part of Assumption 2.3 by showing that features are orthogonal, does so in a toy setting (see previous weakness) and thus fails to demonstrate that these assumptions hold in real-world cases.

Minor weaknesses (did not affect my overall evaluation):

3. Figures 2 and 5 reference Theorem 4, which does not appear anywhere in the paper; this is likely a typo, and the authors probably meant Theorem 3.

4. The distinction between $M_K^{(\overline{K})}$ and $M_{\overline{K}}$ is unclear. The authors define $M_K^{(\overline{K})}$ as the first $\overline{K}$ columns of the $K$-ETF matrix $M_K$; however, it is not evident how this differs from $M_{\overline{K}}$.

**Questions:**

Does the theory provided by the authors also translate into more practical settings and use cases, for instance those in the Model Merging Literature? Especially considering the finetuning operations (finetuned backbone, in many cases with LoRA) and datasets.

---

> ### Author Response · Authors · 2025-11-26
>
> We thank the reviewers for the comments. Please see our general responses which address your concerns in **Weakness 1,2**.
>
> **Weakness 3** This is a typo, and we will fix it in the revised paper.
>
> **Weakness 4** The difference is that $M_{\bar K}$ is a $\bar K\times \bar K$ matrix, and $M_K^{\bar K}$ is a $K\times \bar K$ matrix

---

### Official Review · Reviewer_F1QV · 2025-10-30

**Soundness:** 2
**Presentation:** 2
**Contribution:** 2
**Rating:** 4
**Confidence:** 4

**Summary:**

The paper provides a theoretical analysis of LoRA under a multiclass linear classification setting to explain two empirical observations: its resistance to catastrophic forgetting and the success of adapter merging. The authors show that under certain regularization regimes, the LoRA solution aligns with the hard-margin SVM direction, and derive margin-based expressions to justify these behaviors. Limited experiments on linear-probe settings support the analysis.

**Strengths:**

1. Addresses two important empirical properties of LoRA with an explicit mathematical formulation.
2. Clear presentation and rigorous derivations under simplified assumptions.
3. Offers interpretable margin-based insights and closed-form solutions for merging coefficients.

**Weaknesses:**

1. The theoretical foundation is too shallow, restricted to a single linear layer without deeper or nonlinear analysis, which weakens its relevance to real LoRA behavior in large models.
2. The experimental validation is limited and cannot convincingly demonstrate how LoRA reduces forgetting or benefits merging in practice.
3. The regularization analysis is intuitive rather than novel; showing that a stronger regularizer constrains updates is already well understood.
4. Strong orthogonality assumptions further reduce practical applicability.

**Questions:**

Does the proposed analysis or conclusion still hold when LoRA is applied across all layers or on larger-scale models?

---

> ### Author Response · Authors · 2025-11-25
>
> We thank the reviewers for the comments. Please see the general responses where we answer all your comments.

---

> > ### Comment · Reviewer_F1QV · 2025-11-26
> >
> > Thanks for your responses, after careful consideration, I decided to keep my current score.

---

### Official Review · Reviewer_odn8 · 2025-10-31

**Soundness:** 3
**Presentation:** 3
**Contribution:** 2
**Rating:** 4
**Confidence:** 3

**Summary:**

The paper presents a theoretical study of LoRA in a simplified multi-class linear classification setting. Under strong orthogonality and separability assumptions, it shows that, with appropriate regularization, LoRA can remain resistant to catastrophic forgetting, and that independently trained LoRA adapters can be merged into a single model that maintains good performance across multiple tasks.

**Strengths:**

- The analysis is clear and logically structured.
- The margin-based analysis yields closed-form expressions for both the optimal regularization parameter and the adapter-merging coefficients.
- The paper further distinguishes different penalty regimes to reveal the trade-off between preserving the pre-trained task and fitting the new task.

**Weaknesses:**

- The analysis relies on a number of very strong and idealized assumptions. These assumptions are unlikely to hold under realistic data distributions, and the paper currently does not provide large-scale empirical evidence to indicate how robust the theory is once these assumptions are relaxed.
- The paper only treats a simple multi-class linear classification head with frozen features, whereas in practice LoRA is mostly used inside attention blocks on non-orthogonal, high-dimensional representations; it would be helpful to at least discuss how the results might extend to that setting.
- The experimental section is relatively light and limited to small vision-style settings.

**Questions:**

Could the authors analyze how the current conclusions change when the orthogonality and related assumptions are relaxed to a more realistic, approximately orthogonal setting?

---

> ### Author Response · Authors · 2025-11-25
>
> Thank you very much for your reviews and comments. Please see the general responses for your concerns raised in **Weakness**.
>
> **Q1** We do not currently have theoretical results for the setting where the orthogonality assumptions are relaxed. However, our empirical findings indicate that when the features are approximately orthogonal, the theoretically derived optimal regularization parameter and mixing coefficients still agree closely with the empirical optima, suggesting that the results may continue to hold in this broader regime.

---

### Author Response · Authors · 2025-11-21
**General responses to all reviewers**

We appreciate the reviewers’ concerns regarding the linear setting and the restrictive assumptions. Below we clarify our contributions and explain why these modeling choices are both standard and necessary for developing the **first theoretical** foundations of LoRA adapter merging and the choices of regularization parameters. We respectfully note that simplified assumptions are common and often essential in the deep learning theory community; evaluating theory work primarily based on the generality of its assumptions rather than the insights it provides would rule out much of the progress the community has made in understanding modern neural networks.

**(1) Restatement of our contributions:** The paper makes two primary *theoretical* contributions:

- **Characterizing the implicit bias of LoRA.**
We provide the first characterization of the *stationary points* of LoRA under weight decay. This reveals the structure of the solutions induced by LoRA’s low rank parameterization and explains how LoRA adapts the pre-trained model to the fine-tuning tasks.

- **Using the implicit bias to derive optimal regularization and optimal merging coefficients.** Based on the closed-form characterization of the limiting solution, we analytically determine how to optimally choose the regularization parameter during fine-tuning and how to set the mixing coefficients for adapter merging.  To our knowledge, no prior work provides theoretical guidance on these choices.

Thus, our analysis characterizes the exact optimality conditions and stationary solutions, and uses them to derive principled guidelines for LoRA training and merging.

**(2) Why linear models and orthogonal data? This is standard in deep learning theory**

We acknowledge that the orthogonal-data assumption is idealized. However, similar simplifying assumptions are widely adopted in theoretical deep learning to isolate core mechanisms that would otherwise be analytically intractable. Representative examples include:

- **Neural collapse theory**  Despite neural collapse phenomenon has been observed for deep neural networks, existing theoretical works [1,2,3,4,5] study a *surrogate model* (a matrix factorization objective with regularization) to explain the neural collapse theory.

- **Implicit bias of gradient descent** Classical results on the implicit bias of neural networks [6]  begin with *linear binary classification problems* under linear separable assumptions before extending to more complex settings, such as multi-class linear classification problems [7,8].

- **Theoretical studies of LoRA** Recent theory on LoRA follows the same pattern.  [9] analyze LoRA initialization in the context of matrix factorization;  [10] study gradient flow learning dynamics in matrix factorization;  [11] analyze the LoRA loss landscape for linearized deep networks (effectively linear models);  [12] study one-step LoRA updates for one-layer models;  [13] analyze two-layer ReLU networks under orthogonal features and rank-one assumptions.

Our assumptions are no stronger than those used in prior theoretical work and serve the same purpose: enabling exact, theoretical characterization of LoRA, which is not feasible for general correlated data or deep nonlinear networks.

**(3) Future work:** Relaxing orthogonality and extending the framework to nonlinear features are important next steps. Our work provides the theoretical foundation needed for such generalization, and we view it as a necessary first step toward a deeper understanding of LoRA in modern deep architectures.

[1] Prevalence of neural collapse during the terminal phase of deep learning training

[2] Neural Collapse for Cross-entropy Class-Imbalanced Learning with Unconstrained ReLU Features Model

[3] Imbalance trouble: Revisiting neural-collapse geometry

[4] A geometric analysis of neural collapse with unconstrained features

[5] An Unconstrained Layer-Peeled Perspective on Neural Collapse

[6] The implicit bias of gradient descent on separable data

[7] The implicit bias of gradient descent on separable multiclass data

[8] Unified binary and multiclass margin-based classification

[9] On the crucial role of initialization for matrix factorization

[10] Understanding the learning dynamics of lora: A gradient flow perspective on low-rank adaptation in matrix factorization

[11] Lora training in the ntk regime has no spurious local minima.

[12] Lora-one: One-step full gradient could suffice for fine-tuning large language models, provably and efficiently

[13] Gradient dynamics for low-rank fine-tuning beyond kernels

---

### Author Response · Authors · 2025-11-21
**General Response to Weak Experiment Setup**

**Regarding the experimental setup**
Reviewers noted that our experiments apply LoRA only to the final linear classifier of a pretrained model, rather than to multiple layers. We chose this setting intentionally, for the following reasons:

- **The experiments are designed to evaluate the theory, not benchmark LoRA.** Our theoretical results are derived for a linear model with orthogonal features. The experimental goal is therefore to test whether the *theoretical predictions remain valid when the assumptions are violated*, i.e., on real, (nearly) orthogonal data with realistic feature representations for linear classifier. This is why we use pretrained features with a linear model. Since in the feature space, the data from different classes are nearly orthogonal (see Figure 3 and Figure 4).

- **The results support the robustness of the theory.** Even though real datasets do not satisfy the orthogonal data assumption, our empirical results show that the theoretically predicted optimal regularization parameters and optimal merging coefficients still qualitatively match the empirical optima. This is precisely the type of validation the theory aims for.

Thus, the experimental choices are *deliberate* and aligned with the purpose of the paper: to examine whether our theoretical insights remain predictive in practical scenarios. Extending the framework to multi-layer LoRA is an important but separate direction, and falls outside the scope of the theoretical model studied here.

---

### Author Response · Authors · 2025-11-25
**General response to Extension to Apply LoRA to Multi-layer**

Response: We provide an initial answer to this question under the neural tangent kernel (NTK) regime, suggesting that our framework can also explain the use of adapters in the backbone.

In [1], the authors empirically verify that, under certain conditions, LoRA fine-tuning operates in the NTK regime, where the LoRA adapter weights remain close to their initialization. In this regime, our analysis can be extended from linear probing to the general case where LoRA is applied to backbone models, following similar reasoning to that in [2].

Specifically, we consider a pre-trained deep network parametrized by $\Theta_{\mathrm{pre}}=\lbrace\theta_1,\ldots,\theta_L\rbrace$ and apply LoRA to each layer with low-rank adapters $\lbrace B_{i},A_{i}\rbrace_{i=1}^{L}$. The fine-tuned model is given by: $f(x;\lbrace \theta_i+B_{i}A_{i}\rbrace_{i=1}^L)$.

In the NTK regime, we linearize the model around the pre-trained parameters as:

$$ f(x; \lbrace\theta_i + B_i A_i\rbrace_{i=1}^L)\approx f(x; \lbrace\theta_i\rbrace_{i=1}^L) + \sum_{i=1}^L \left\langle \nabla_{\theta_i} f(x; \lbrace\theta_i\rbrace_{i=1}^L), B_i A_i \right\rangle. $$

In the fine-tuning setting, we receive a dataset $\mathcal{D}_{\mathrm{ft}}=\lbrace (\bar{x}_i, \bar{y}_i)\rbrace_{i=1}^{n_{\mathrm{ft}}}$. To simplify the expression and make the residual term dominant, we make the following assumption:

Assumption $2.2^{\ast}$: The pre-trained model outputs zero on the fine-tuning data, i.e., $f(\bar{x}_i; \lbrace\theta_i\rbrace_{i=1}^L) = 0$ for all $i \in [n_{\mathrm{ft}}]$.

Under this assumption, the model simplifies to:

$$ f(x; \lbrace\theta_i + B_i A_i\rbrace_{i=1}^L) \approx \sum_{i=1}^L \left\langle \nabla_{\theta_i} f(x; \lbrace\theta_i\rbrace_{i=1}^L), B_i A_i \right\rangle. $$

To further simplify, we define the following notations:

$$ U = \begin{bmatrix}B_1 \\ B_2 \\ ...\\ B_L \end{bmatrix},\quad V = \begin{bmatrix}A_1^\top \\ A_2^\top \\ ... \\ A_L^\top \end{bmatrix},\quad G(x)=\mathrm{diag}\bigl(\nabla_{\theta_{1}} f(x; \lbrace\theta_{i}\rbrace_{i=1}^L), ..., \nabla_{\theta_L} f(x; \lbrace\theta_{i}\rbrace_{i=1}^L)\bigr). $$

Then, the model output can be expressed as:

$$ f(x; \lbrace\theta_{i} + B_{i} A_{i}\rbrace_{i=1}^L) \approx \left\langle UV^\top, G(x) \right\rangle. $$

This expression takes the same form as linear probing, except that the feature representations are now given by $G(x)$, which are the neural tangent features of the backbone model. Consequently, all results from our linear classification setting directly apply to this NTK-linearized formulation.

==================================================

Reference

[1] Malladi, Sadhika, et al. "A kernel-based view of language model fine-tuning." International Conference on Machine Learning. PMLR, 2023.

[2] Jang, Uijeong, Jason D. Lee, and Ernest K. Ryu. "LoRA training in the NTK regime has no spurious local minima." arXiv preprint arXiv:2402.11867 (2024).

---

### Meta-Review · Area_Chair_Ma6M · 2025-12-19

**Summary:**

Initial concerns centered on the paper’s reliance on strong orthogonality and linear model assumptions, and an experimental setup that applies LoRA only to the final linear classifier, which several reviewers viewed as too far from practical multi layer LoRA usage. Reviewers also requested clearer evidence that the theoretical prescriptions for optimal regularization and adapter merging remain predictive when assumptions are relaxed, and asked for broader baselines and more realistic evaluations. The authors addressed these by clarifying the paper’s scope as theory first rather than benchmarking, arguing that pretrained feature spaces are approximately orthogonal in their settings and that the empirically optimal regularization and merging coefficients qualitatively match theoretical predictions. They additionally provided an NTK regime argument indicating how the linear probing analysis can extend to multi layer LoRA via NTK linearization, and acknowledged that fully relaxing orthogonality is an important open direction. Overall, reviewers agreed the derivations are clean and the margin based closed forms are interpretable, but the panel remained mixed because the empirical validation under realistic conditions is still limited and two reviewers remained strongly negative.

**Reviewer Concerns:**

The reviewers initially raised several key concerns:

(1) strong and idealized assumptions, especially orthogonality and separability, with unclear robustness beyond toy or near orthogonal regimes;

(2) limited experimental validation and an unrealistic fine tuning setting that only adapts the classifier head, with missing or weak links to common multi layer LoRA use in attention blocks;

(3) insufficient evidence for practical relevance to LLMs and model merging practice, and requests for stronger comparisons and broader evaluations;

(4) clarity issues in presentation, including minor typos and definition distinctions noted by reviewers.

**Reviewer Scores:**

Scores remained split, with multiple borderline reviews at 4 and at least one reject at 2, reflecting appreciation for the mathematical completeness but dissatisfaction with the experimental coverage and practical alignment. Reviewers odn8 and F1QV acknowledged clear derivations and useful closed form expressions but maintained below threshold ratings due to limited robustness evidence and narrow experiments, and F1QV explicitly kept their score after the author response. Reviewers Qs8n and 85Bk were more critical, emphasizing that linear probing with frozen backbones does not validate claims about LoRA behavior in realistic multi layer fine tuning, and that the strong assumptions are insufficiently justified empirically. The author responses improved clarity on intent and provided a plausible NTK based extension argument, but did not fully close the gap between the simplified theory and practical LoRA deployments in the eyes of the panel.

---

### Decision · Program_Chairs · 2026-01-26

Reject